# The chromatin landscape of healthy and injured cell types in the human kidney

There is a need to define regions of gene activation or repression that control human kidney cells in states of health, injury, and repair to understand the molecular pathogenesis of kidney disease and design therapeutic strategies. Comprehensive integration of gene expression with epigenetic features that define regulatory elements remains a significant challenge. We measure dual single nucleus RNA expression and chromatin accessibility, DNA methylation, and H3K27ac, H3K4me1, H3K4me3, and H3K27me3 histone modifications to decipher the chromatin landscape and gene regulation of the kidney in reference and adaptive injury states. We establish a spatially-anchored epigenomic atlas to define the kidney's active, silent, and regulatory accessible chromatin regions across the genome. Using this atlas, we note distinct control of adaptive injury in different epithelial cell types. A proximal tubule cell transcription factor network of *ELF3*, *KLF6*, and *KLF10* regulates the transition between health and injury, while in thick ascending limb cells this transition is regulated by *NR2F1*. Further, combined perturbation of *ELF3*, *KLF6*, and *KLF10* distinguishes two adaptive proximal tubular cell subtypes, one of which manifested a repair trajectory after knockout. This atlas will serve as a foundation to facilitate targeted cell-specific therapeutics by reprogramming gene regulatory networks.

The cellular response to kidney injury causes gene expression changes, leading to successful repair with restored function or failed tubular epithelial repair that can progress to chronic kidney disease (CKD)[1,2]. Putative adaptive states with successful or maladaptive characteristics were recently described[3] in both the proximal tubule (PT) and the thick ascending limb (TAL) epithelial cells of the loop of Henle. Gene expression profiles found in these adaptive states included epithelial to mesenchymal transition, elevation of cytokine production, senescence, and downregulation of ion and solute transporters required for normal physiological function of these nephron segments. Expression of key injury, repair, and progenitor markers were altered in these states[3–5]. The proportion of adaptive cell state signature was increased in individuals with CKD or acute kidney injury (AKI) and correlated with faster progression to end-stage renal disease[3]. Although these studies provide important insights into gene expression changes in kidney injury and repair, a critical need persists to define the

epigenetic regulation of gene expression as kidney cells progress through states of health, injury, regeneration, or failed repair.

Activation or repression of gene transcription is regulated by changes in the epigenetic landscape, including the sliding of nucleosomes to create regions of open and closed chromatin, DNA methylation, and histone modification[6]. Numerous investigations have uncovered these epigenetic signals in kidney disease[7–9]. Different patterns of DNA methylation have been found in DKD with alterations in tumor necrosis factor-alpha and other pro-fibrotic genes[10–12]. Putative regulatory regions have been identified in diabetic kidney disease (DKD), renal cell carcinoma (RCC), and polycystic kidney disease (PKD)[11,13,14] using ATAC-seq to determine the accessibility of chromatin across the genome. A broad set of ATAC-seq data is now available for a variety of renal diseases, including RCC and DKD[11,13,15], and has also been leveraged to infer tissue-specific causal regulation of GWAS findings[16]. The snATAC-seq of kidney tissue has augmented our

✉ e-mail: sanjayjain@wustl.edu; mrauchma@wustl.edu; meadon@iupui.edu

understanding of the regulation of disease at the single-cell level. In one study, reduced accessibility of glucocorticoid receptor binding sites was observed in the proximal tubule of diabetic humans[15]. Investigations have also examined chromatin accessibility in human Papillary RCC (pRCC), kidney organoid differentiation[5,17,18], and in multiple murine models[19,20]. Each of these studies provides important insights into epigenetic expression regulation in the kidney. However, deeper insights would be gained by the integration of these technologies, particularly with single-cell sequencing, spatial transcriptomics, and sequencing to delineate histone modifications.

To accurately identify regulatory regions that orchestrate the coordinated activation and silencing of gene networks in injury and repair, we interrogated an overlapping set of human kidney tissue samples with multiple technologies to establish an atlas of genome-wide DNA methylation, histone modifications associated with active or repressed regions in promoters and enhancers, open chromatin, gene expression, and protein expression. These methods have undergone rigorous QA/QC review and standardization[21], and have been optimized for relatively low cell inputs suitable for interrogation of kidney biopsy samples. Orthogonal validation was attained at single-cell resolution and with spatial localization. We identified differential regulation of the adaptive cell state in the PT and TAL. The kidney epigenomic atlas established here will prove a valuable reference for future studies analyzing diseased tissue obtained from biopsy specimens.

## Results

### Samples and quality control

To comprehensively understand the contributions of epigenomic features to transcript expression regulation, we generated, integrated, and aligned data from multiple orthogonal technologies performed on a partially overlapping set of 25 unique specimens (including 23 unique epigenetic samples, Fig. 1, Supplementary Data 1). Multiple replicates and multiple technologies were run for some samples, resulting in a total of 84 datasets. Epigenetic assays used included: (1) WGBS from laser microdissected glomeruli (GLOM) and tubulointerstitium (TI), (2) single nucleus multiome (10x Genomics combined snATAC-seq and snRNAseq), (3) bulk ATAC-seq, and (4) CUT&RUN for histone

modifications of active chromatin (H3K27ac, H3K4me3, H3K4me1) and repressive chromatin (H3K27me3) derived from kidney cortex. A kidney chromatin landscape browser application was developed to view these aligned datasets across the genome (https://doi.org/10.48698/HHE6-YV15). Spatial transcriptomic, regional transcriptomic, and regional proteomic expression datasets complemented the epigenetic technologies.

We established extensive quality control assessments for the generated datasets to determine technical and biological reproducibility, coverage, batch effect, and assay drift. For WGBS, samples clustered as expected based on the region dissected (glomeruli and the tubulointerstitium) with an average coverage of 25×, and a mean of 22,156,845 CpG sites (range: 19.9–25.7 million sites) mapped per sample with calculated methylation, representing 81% of all CpG sites and >99% of annotated CpG islands within the human genome hg38 (Supplementary Fig. 1). The CUT&RUN data was highly reproducible and showed strong correlation across samples (Supplementary Fig. 2) and overlap with the ENCODE ChIP-seq and DNAse-seq datasets, providing additional validation of our approach in human kidney (Supplementary Fig. 3). The multiome dataset also showed robust analytical and biological reproducibility with ~2400 average genes/cell detected, expected cell type distribution, and 4.8 mean transcriptional start sites (TSS) enriched per cell (Supplementary Fig. 4).

### Integrated whole epigenome alignment

After demonstrating satisfactory data quality, we examined the genome-wide alignment of the open chromatin signatures of the whole kidney by first using a publicly available ATAC-seq dataset[22] with histone marks and methylation patterns (Fig. 2). These modalities aligned across the genome. For example, for *PODXL*, a gene expressed in podocytes, there was a strong methylation dip (reduction or absence of DNA methylation) that coincided with an ATAC-seq open chromatin peak, and the H3K4me3 promoter mark near the TSS (Fig. 2a). Furthermore, two intronic ATAC-seq peaks and a peak in the final exonic region aligned with H3K27ac active chromatin marks, H3K4me1 enhancer marks, and glomerular methylation dips. This multimodal approach allowed more detailed insights into mechanisms and gene regulatory regions that define *PODXL* expression. We also

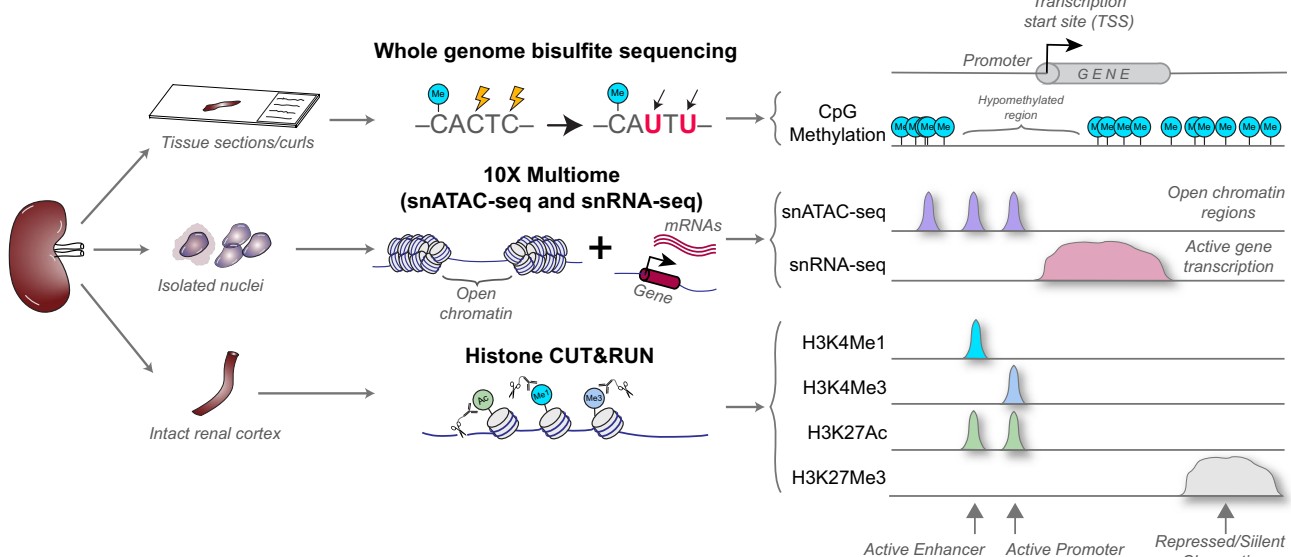

**Fig. 1 | Study workflow.** In an overlapping set of kidney samples, tissue was interrogated by laser microdissection-guided whole genome bisulfite sequencing (WGBS), by multiome single nucleus Assay for Transposase-Accessible Chromatin for sequencing (snATAC-seq) and single nucleus RNA sequencing (snRNAseq) after cell disaggregation, and by Cleavage Under Targets & Release Using Nuclease (CUT&RUN) of kidney cortex. Datasets were aligned in the human genome 38 to create an integrated epigenomic atlas.

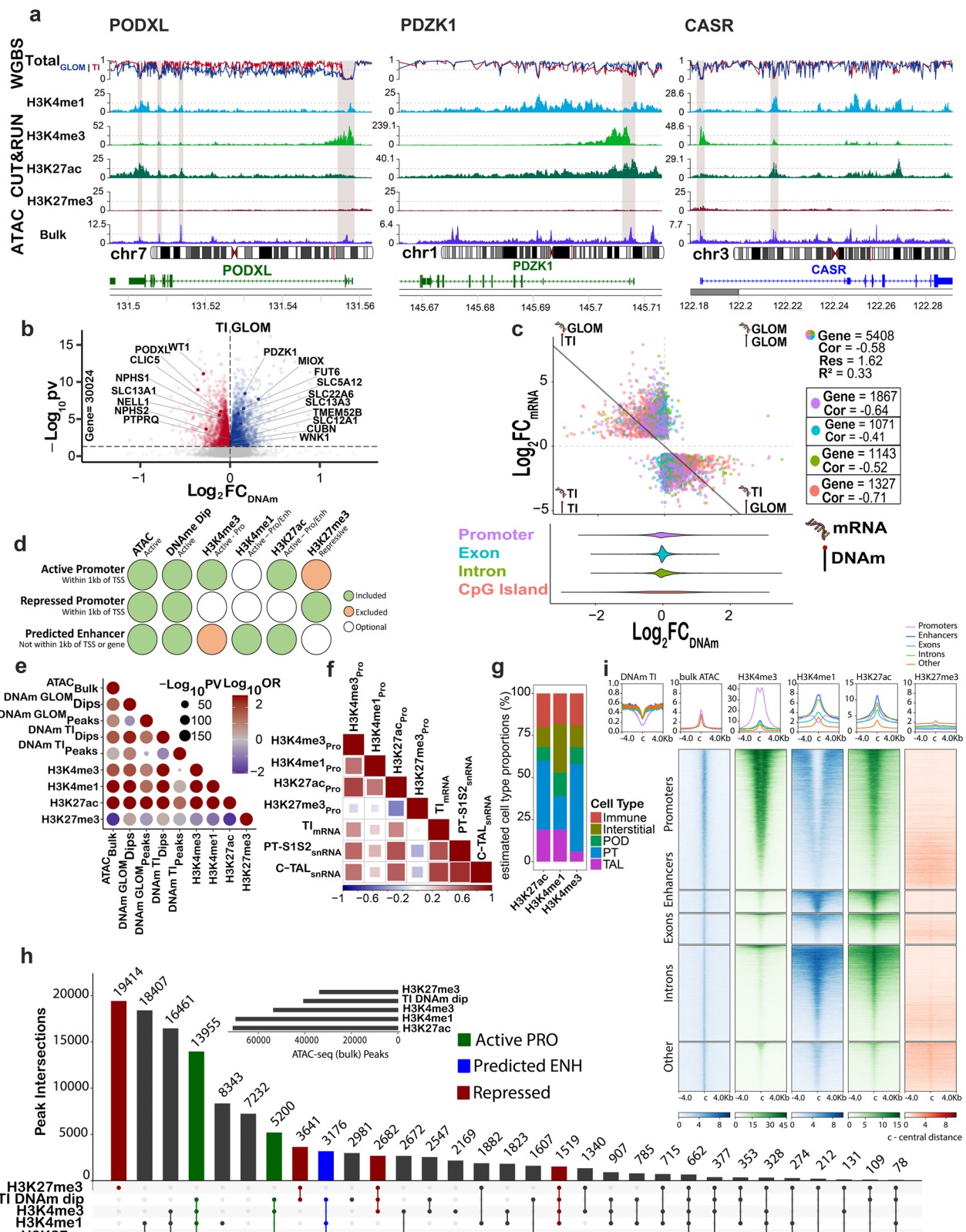

confirmed this correspondence in epigenetic regulation for proximal tubule and TAL-specific genes *PDZK1* and *CASR*, respectively.

We next assessed regulation in key kidney structures by first examining differential methylation patterns between glomeruli and the tubulointerstitium (Supplementary Data 2). As expected, we observed an inverse relationship between gene expression and methylation in these structures; genes expressed in glomeruli (e.g.,

*WT1*) were methylated in tubules and tubule-expressed genes (*MIOX*) were methylated in glomeruli (Fig. 2b). Similar trends were observed for promoter methylation, summative intronic methylation, exonic methylation, CpG island methylation, and methylation in the 5' and 3' untranslated regions (UTR); however, genes were not differentially methylated to the same extent in these gene regions (Supplementary Fig. 1n). Given the variability in differential gene region methylation,

**Fig. 2 | Alignment of epigenomic features in bulk and regional human kidney samples. a** Epigenomic features of marker genes for glomerulus (*PODXL*), proximal tubule (PT-S12, *PDZK1*), and thick ascending loop of Henle (C-TAL, *CASR*), displaying: (1) DNA methylation (DNAm) in the tubulointerstitium (TI) (*N* = 15) and glomerulus (GLOM) (*N* = 15), (2) bulk CUT&RUN for four histone modifications: H3K27ac (*N* = 6), H3K27me3 (*N* = 10), H3K4me1 (*N* = 3), H3K4me3 (*N* = 3), and (3) assay for transposase-accessible chromatin using sequencing (ATAC-seq) on bulk tissue (Encode ENCSR297VGU) (*N* = 1). Gray stripes indicate active promoters wherein ATAC-seq peaks at transcriptional start sites coincide with DNAm dips, and H3K4me3 peaks. Variable H3K27ac peaks reflect compartments' proportion within bulk tissue. **b** Differential methylation between GLOM and TI kidney compartments for summative methylation of 30,024 gene bodies *P* value < 0.05 by *t* test. **c** Best-fit regression model of methylation and mRNA expression in identical samples (*N* = 22) for differentially expressed genes (*N* = 5408) between the GLOM and TI. Each dot represents a gene. *Y* axis is the Log₂ fold change of mRNA between the GLOM and TI. *X* axis is the Log₂ fold change of methylation between the GLOM and TI. The best-fit annotated gene region (summative promoter, exon, intron, of CpG island methylation) with the most negative correlation was identified as the promoter for 1867 genes and CpG island for 1327 genes. The inset represents methylation fold change distribution in annotated gene regions. **d** Genomic region annotation criteria based on epigenetic landscape. **e** Landscape correlation agreement between datasets (Fisher's exact test two sides). **f** Histone markers and spearman correlation with snRNAseq expression in the PT-S12 and C-TAL and in the regional mRNA expression of the microdissected TI. **g** Cell type deconvolution of CUT&RUN for H3K27ac, H3K4me1, H3K4me3 active histone modifications at promoters. The RNA signature was taken from the HuBMAP/KPMP atlas snRNAseq (*N* = 36), using the 10% most DE marker genes. **h** Upset plot depicting overlap in peaks of H3K27ac, H3K4me1, H3K4me3, and H3K27me3 with DNAm dips across the genome. Active promoters, predicted enhancers, and repressed regions are annotated. **i** Heatmap of CUT&RUN marks across the genome by annotated region after filtering for open chromatin and DNA methylation dips. The figure uses a licensed stock image adapted from Adobe Illustrator (Eadon et al. · stock.adobe.com).

we sought to determine the key regulatory regions where methylation contributes to differential expression within the kidney. To accomplish this, we acquired mRNA and protein expression (by mass spectrometry) from the same microdissected samples as those interrogated with WGBS (glomeruli and tubulointerstitium). Univariate comparisons of expression with summative gene region methylation were performed for all differentially expressed genes (DEGs, *n* = 5408) and proteins (DEPs, *n* = 1796) between the glomeruli and tubulointerstitium (Supplementary Fig. 5). We applied a best-fit model (overall *C* = 0.58) of expression and methylation across all DEGs including the region with the highest c-statistic (a measure of model fit) for each individual gene (Fig. 2c). Summative promoter region methylation best-explained mRNA expression in 1867 of the 5408 DEGs, while CpG island methylation best-explained expression in 1327 genes. Albeit non-canonical, intronic (*n* = 1143) and exonic (*n* = 1071) methylation best correlated with expression in a subset of DEGs, but these genes had a lower overall correlation between methylation and expression. Similar correlation patterns were observed in methylation and protein expression regressions (Supplementary Fig. 5). We hypothesized that summative methylation from regions defined by CUT&RUN histone mark peaks might also correlate with mRNA expression. The strongest correlation between methylation and mRNA was found for regions in which methylation was defined by enhancer H3K4me1 peaks (*C* = 0.45), but this did not outperform the best-fit model of Fig. 2c. Even in the best-fit model, the correlation was 0.58, suggesting features beyond methylation contribute significantly to expression regulation.

We assayed genome-wide DNA methylation coexistence with CUT&RUN histone marks in the kidney cortex and bulk kidney ATAC-seq peaks. Across the genome, open chromatin regions were positively associated with active chromatin marks (H3K27ac, H3K4me1, H3K4me3), and TI methylation dips, whereas the silencing histone mark (H3K27me3) displayed a negative association with these features (Fig. 2d, e). These patterns suggest conserved features of epigenomic regulation across the genome. Epigenomic features at gene promoters were then compared with mRNA expression using laser microdissection-guided tubulointerstitial expression[23] or pseudobulk snRNAseq expression of the PT (S1 and S2) and cortical TAL (C-TAL)[3]. The active chromatin marks (H3K4me3, H3K4me1, and H3K27ac) positively correlated with expression, while silencing marks were negatively correlated (Fig. 2f). We justified these comparisons because the most abundant cell types of bulk tissue were the PT and cortical TAL cells and the tubulointerstitium accounts for the vast majority of these cell types. We performed cell type deconvolution of the histone modifications within promoters and identified major contributions from the PT, TAL, and interstitial cell type signatures within the cortex histone data (Fig. 2g). Similarly, bulk ATAC-seq peaks display the strongest association with active promoters and predicted enhancers associated with PT specific genes (Supplementary Fig. 6a). Approximately half of all TI DNA methylation dips overlapped with bulk ATAC-seq peaks, but very few ATAC-seq peaks overlapped in regions of elevated DNA methylation (Supplementary Fig. 6b, c). Further, TI DNA methylation dips that overlapped ATAC-seq peaks showed stronger histone modification patterns than those that lacked overlapping ATAC-seq peaks (Supplementary Fig. 6d). We quantitatively examined the overlap between TI DNA methylation dips with cortex histone modifications at bulk ATAC peaks. We identified a strong overlap at peaks associated with epigenetic patterns indicative of active promoters (H3K4me3+, H3K27ac+, and H3K4me1+/−) but less overlap with peaks displaying active enhancer (H3K4me1+ and H3K27ac+) and repressed region (H3K27me3) histone marks (Fig. 2h). To enhance specificity to TI, bulk ATAC peaks were filtered for those overlapping TI DNA methylation dips and we examined the binding patterns of promoter H3K4me3, enhancer H3K4me1, active H3K27ac, and silencing H3K27me3 histone modifications in distinct genomic regions (Fig. 2i). As expected, H3K4me3 localized strongly to promoters which also display H3K27ac binding, indicating that the majority of the promoters selected by the overlap of bulk ATAC-seq peaks and TI DNA methylation dips are in the active state. H3K4me1 is expectedly localized outside of promoters (within predicted extragenic enhancers, exonic, and intronic peaks) together with H3K27ac, indicating that the majority of the remaining ATAC peaks with DNA methylation dips display active enhancer features. This integrative analysis combining bulk tissue ATAC and cortex CUT&RUN histone data with region-specific DNA methylation demonstrates the ability to use bulk-level data to augment region-specific data and gain additional insights into region-specific epigenetic landscape information.

## Cell-type specific open chromatin in the reference cell state

We leveraged the 10x multiome data to better understand the cell type-specific open chromatin signature across the kidney's diverse range of cell types. The multiome dataset included 47,217 nuclei derived from 12 samples, which partially overlapped with those used in WGBS and CUT&RUN (Fig. 3). The multiome atlas yielded 72 distinct cell types based on a pre-existing human kidney atlas[3]. As a control, we examined the alignment of expression and open chromatin in known marker genes of the podocytes (POD), proximal tubules (PT), and cortical thick ascending loop of Henle (C-TAL) in Fig. 3c. The area under-the-curve (AUC) of the TSS was highest in the POD for *PODXL* and *NPHS1*, while the TSS AUC was highest for *LRP2* and *SLC5A2* in the PT. The regulatory patterns of *PDZK1* (expressed in PT), *ESRRG* (expressed in C-TAL), and *PODXL* (expressed in POD), align across snATAC-seq, WGBS in the GLOM and TI, and the 4 histone modifications (Fig. 3d–f). For example, in *PDZK1*, open chromatin was differentially accessible in the PT (combined S1 and S2) with the TSS peak

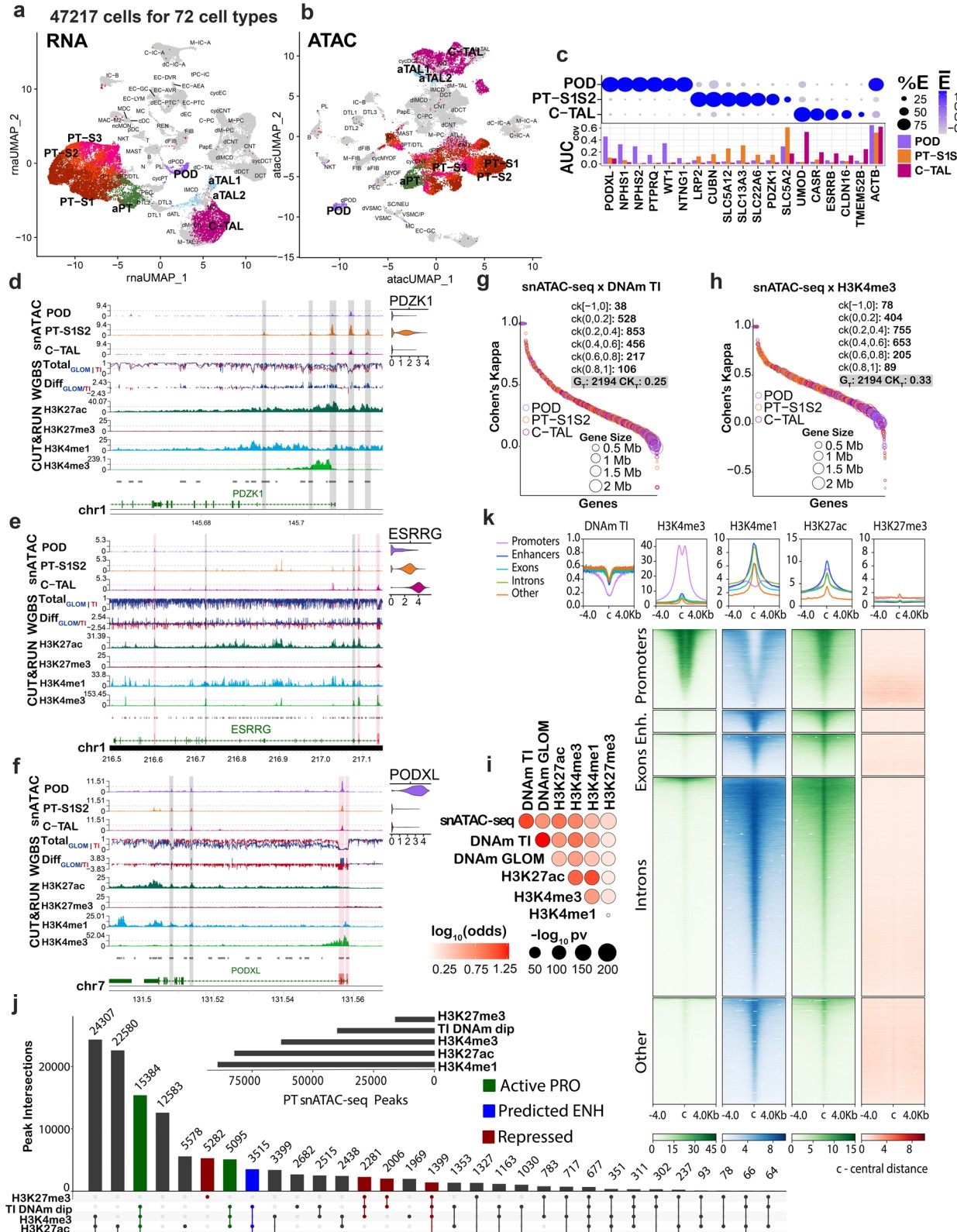

aligning with a TI methyl dip, and a peak in H3K27ac with an H3K4me3 promoter peak in close proximity. Two additional upstream (1–10 Kb) multiome peaks also aligned with methyl dips and H3K27ac peaks, albeit with H3K4me1 enhancer peaks. In *ESRRG*, the first region highlighted in red denoted a CpG island at the TSS with a methyl dip that aligned with open chromatin of only the C-TAL cell type as well as the

H3K4me3 promoter and H3K27ac activation marks. A small peak in H3K27me3 also aligned, likely due to the bulk nature of the CUT&RUN assay.

To confirm the alignment between the snATAC-seq peaks with the other technologies, we selected 2194 DEGs from POD, PT, and C-TAL. The regions of each gene were coded as "peak" and "absent peak"

**Fig. 3 | Single-cell epigenomics in health.** The multiome reduction by uniform manifold approximation and projection (UMAP) of 47,217 nuclei in 72 clusters and 12 samples aligning with the HuBMAP/KPMP atlas is depicted for **a** snRNA-seq and **b** snATAC-seq assays. Highlighted clusters include podocytes (POD), proximal tubule (PT) S1, S2, S3 and adaptive PT (aPT), cortical thick ascending loop of Henle (C-TAL), and adaptive TAL (aTAL1 and aTAL2) cells. **c** Gene markers for POD, PT-S1 merged with S2 (PT-S12), and C-TAL reveal cell type specificity of expression and chromatin accessibility. Dot plot (top) reveals transcript expression. Bar graph (bottom) represents the area under the curve (AUC) for open chromatin coverage summed across the entire gene. Genes **d** *PDZK1*, **e** *ESRRG*, and **f** *PODXL* align across the snATAC-seq, WGBS, and CUT&RUN technologies. snATAC-seq peaks are displayed in tracks 1–3 with respective RNA expression ($N = 12$) in adjacent violin plots. DNAm levels (track 4) and differential DNAm (track 5) are depicted for the GLOM ($N = 15$, blue) and TI ($N = 15$, red). Histone marks for CUT&RUN are found in track 6–9: H3K27ac ($N = 10$), H3K27me3 ($N = 6$), H3K4me1 ($N = 3$) and H3K4me3 ($N = 3$),

respectively. Aggregate open chromatin regions (track 10), and chromosome coordinates are below. Differentially accessible open chromatin peaks are annotated as red stripes (coincides with CpG island) or gray (no CpG island). Cohen's Kappa (CK) agreement between snATAC-seq peaks with DNAm dips (**g**) or H3K4me3 peaks (**h**) in the 2194 differentially expressed genes of POD, PT-S12, and C-TAL. Perfect agreement (disagreement) is 1 (−1), ranging from CK [0,0.2] for no agreement to CK [0.8,1] for perfect agreement. Average CK across all genes: $G_T$. Most peaks positively correlated between technologies. Smaller genes have stronger correlation. **i** Association of DNAm & histone marks with open chromatin. The correlation of open chromatin peaks with DNAm dips, and histone mark peaks is given for differentially expressed genes of the POD, PT-S12, and C-TAL (Fisher's Exact test two sides). **j** Upset plot shows the intersection of CUT&RUN peaks and DNAm dips across snATAC-seq PT-S12/aPT peaks, identifying regulatory regions for PT. **k** Heatmap of CUT&RUN marks in PT-S12/aPT in each annotated region after filtering for open chromatin and DNAm dips.

regions for the snATAC-seq and H3K4me3 datasets, and as "dip" or "absent dip" for TI WGBS. A comparison across technologies using the Cohen's Kappa test was performed to quantitate alignment of peaks across all chromatin of the DEGs (Fig. 3g, h, Supplementary Data 3). Alignment ranged from −1 for perfect disagreement to 1 for total agreement. The agreement was strong between technologies in the DEGs with >99% of genes having a positive Cohen's Kappa value and the strongest agreement was seen in smaller genes. In Fig. 3i, the correlation plot calculated by Fisher's exact test showed agreement between all technologies, with the exception of the H3K27me3 silencing mark, which is expected. These results were in agreement with those from publicly available snATAC-seq data generated by the Humphreys lab[5] (Supplementary Fig. 7b, d). Agreement was also seen with differentially upregulated regions of DNA methylation in the glomerulus and the corresponding expected TI gene expression (Supplementary Fig. 7c), as well as with other technologies (Supplementary Fig. 7e).

We explored the overlap of regulatory regions within the snATAC-seq peaks of the combined PT-S1 and PT-S2 clusters (PT-S12) (Fig. 3j). DNA methyl dips frequently coincided in regions of active promoters, active enhancers, and repressed promoters. Upon filtering the snATAC-seq peaks for regions with DNA methyl dips, peaks of H3K4me3 marks were strongly associated and centered upon known promoter regions. In contrast, H3K4me1 peaks were centered in regions outside of promoters. In both cases, H3K4me3 and H3K4me1 peaks strongly correlated with H3K27ac enrichment but not H3K27me3 (Fig. 3k). This integrative analysis resulted in a PT-S12 specific snATAC-seq peak map with DNA-methylation-and-histone-modification-landscape information for each peak, thereby providing the ability to functionally define each open chromatin region.

We sought to understand the specificity of multiome signals in cell subtypes of the atlas, comparing expression and open chromatin across the PT-S1, PT-S2, and PT-S3 cell clusters. The PT shared expression of common gene markers (*LRP2*, *CUBN*, *SLC5A12*) across all three segments; however, unique gene markers specific to each proximal subsegment were also identified (Supplementary Fig. 7f, g). For example, *PRODH2* was most highly expressed in the PT-S1 with differential accessibility (DA) in its TSS and 3' UTR. *SLC34A1* (Sodium phosphate co-transporter 2a) was highly expressed in the PT-S2, with DA seen in most peaks. Finally, *SLC7A13* (Aspartate/Glutamate Transporter 1) was expressed most highly in the PT-S3 with DA of its TSS and other intragenic regions (Supplementary Fig. 7h–j). Through these analyses, we have demonstrated that the multiome dataset contains the resolution to identify PT subtypes. Additionally, expected open chromatin patterns were seen for the distal convoluted tubule (*SLC12A3*), principal cells (*AQP2*), and endothelial cells (*PECAM1*) in known marker genes (Supplementary Fig. 8).

## Adaptive cell state in the proximal tubule

Recently, important and varied cell states associated with injury have been annotated in a comprehensive atlas of the kidney[3]. One such cell state is the adaptive state, found in multiple epithelial cells of the kidney, including the PT and C-TAL. The adaptive cell state refers to the balance of adaptive repair with maladaptive epithelial to mesenchymal transition. The state is defined by a loss of expression of canonical markers (like *PDZK1* or *SLC5A12* in the PT) with upregulation of injury markers such as *ITGB3*, *PROM1*, and *TPM1*.

We used the multiome data to identify a corresponding set of 4194 DEGs between PT-S12 and the aPT, in alignment with that of the published kidney atlas[3] (Fig. 4a, Supplementary Data 4). The diffusion map (Fig. 4b) reveals the developmental trajectory of the PT, from PT-S12 to the adaptive PT (aPT) cell state with feature plots indicating the loss of canonical genes and upregulation of injury markers over the pseudotime transition to the aPT phenotype (Fig. 4c).

We observed 137,993 peaks (labeled from 1 to 137,993 with gene name and peak number) within the whole genome of the PT-S12 and aPT clusters (Supplementary Data 5). Upon annotation, 21,790 peaks were mapped to within 5 kb of 2 or more genes. From the annotated peaks, 49,662 coincided with a TI DNA methylation dip (Fig. 4d). The differential accessibility was queried between all peaks in the PT-S12 and aPT genes, revealing 10,506 as DA, with 3,112 occurring in promoter regions. In Fig. 4e, 27,174 peaks coinciding with DNA methyl dips were categorized as active promoters (Act-Pro), predicted enhancers (Pred-Enh), and repressed promoters (Repr-Pro) based on histone landscape information and proximity to the TSS. Of these, 6607 were DA peaks. PT peaks aligning with a DNA methyl dip were more likely to be found in a CpG island (Supplementary Fig. 9). PT peaks aligning with enhancer and repressed promoter marks, but not active promoter marks, were more likely to occur in CpG islands. The relationship was maintained after filtering for DNA methyl dips (Supplementary Fig. 9a).

Transcription factors (TF) are proteins that bind to specific DNA sequences and regulate gene transcription by activating or repressing expression. We identified the transcriptional pathways that are associated with differentially regulated regions in chromatin in aPT cells using the TRIPOD package[24]. This permitted the identification of regulatory trios of TF expression, target gene expression, and target gene DA in a multiomic dataset (Supplementary Data 6). If a TF DNA binding motif is enriched at a particular peak, this suggests that the TF is likely regulating target gene expression. If a particular histone modification is enriched, it suggests that the modification is involved in regulating the accessibility of the target gene chromatin. We analyzed 96 DEGs meeting all criteria (DA, DEGs, TF binding in TRIPOD, DNA methyl dip, histone modification present), with 39 DEGs upregulated in PT-S12 and 57 DEGs upregulated in the aPT. The analysis revealed 68 distinct pathways, organized into 40 clusters (Supplementary Fig. 9e,

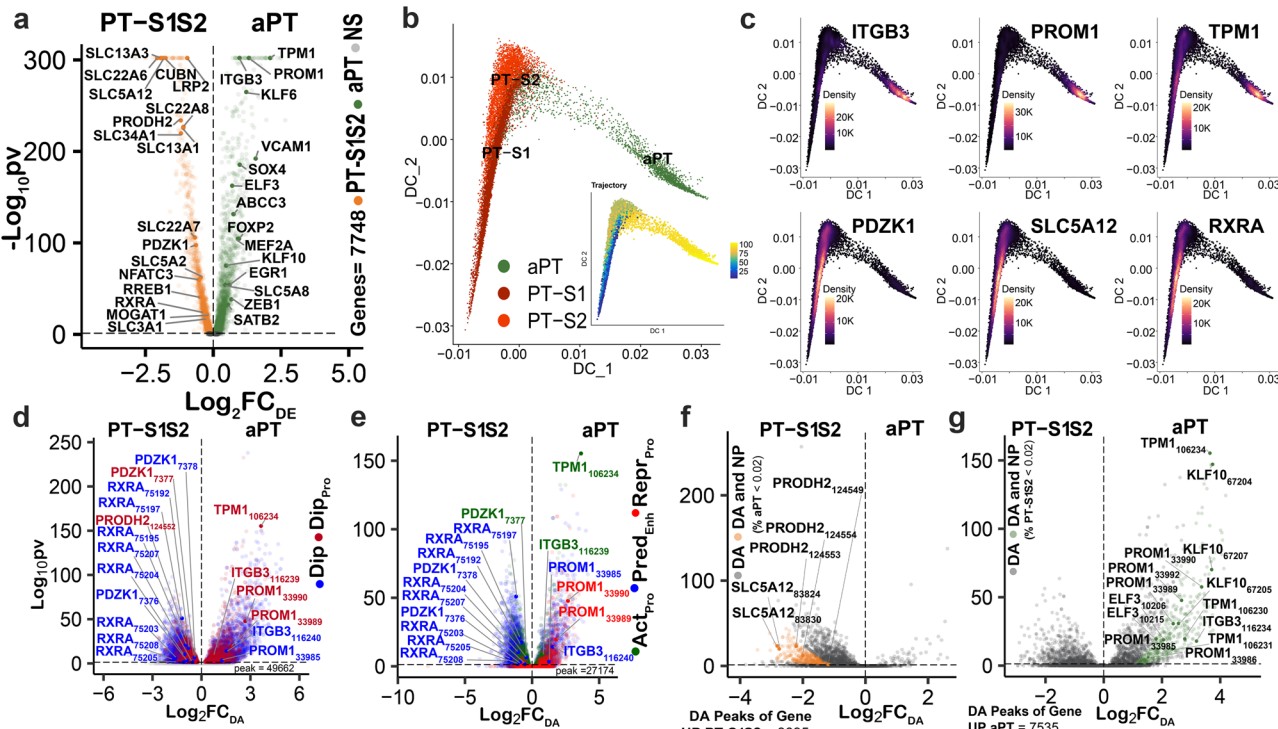

**Fig. 4 | Adaptive cell state in the proximal tubule. a** Differentially expressed genes (DEGs) between the PT-S12 and aPT cell types within the multiome atlas ($N = 12$), 4194 genes with Bonferroni-adjusted $P$ value < 0.05 (Wilcox test). **b** Diffusion map of PT-S1, PT-S2 and aPT with 13,241 nuclei. The inset shows the pseudotime trajectory from PT to aPT. **c** Gene expression localization in aPT cells (*ITGB3, PROM1, TPM1*) for aPT marker genes. Canonical PT markers (*PDZK1, SLC5A12,* and *RXRA*) localize to the PT-S12. **d** Differentially accessible (DA) peaks $N = 10,506$, Bonferroni-adjusted $P$ value < 0.05 (LR test) between the PT-S12 and aPT (from multiome TRIPOD-seurat-aPTxPT-S12 object that coincide with TI DNA methylation (DNAm) dips ($N = 15$). Red = promoter dip, blue = dip outside

promoter. **e** DA multiome peaks with DNAm dips and a CUT&RUN histone mark peak ($N = 22$). Active promoter (Act$_{pro}$) = green, predicted enhancer (Pred$_{enh}$) = blue, repressed promoter (Rep$_{pro}$) = red, $N = 6607$, Bonferroni-adjusted $P$ value < 0.05 (LR test). **f** DA peaks ($N = 2557$) in upregulated DEGs of the PT-S12, targeted by a transcription factor (TF) in TRIPOD analyses. Orange dots represent new peaks (NP) in the PT-S12, where fewer than 2% of aPT nuclei had open chromatin Bonferroni-adjusted $P$ value < 0.05, LR test). Gray dots display DA peaks present in both PT-S12 and aPT. All DA NP were found in PT-S12 cells for DEGs upregulated in the PT-S12. **g** DA peaks ($N = 4573$) in upregulated DEGs of the aPT. Green = aPT NP. Gray = DA peak present in both PT-S12 and aPT.

Supplementary Data 7), with the top cluster being platelet-derived growth factor binding, which is important in myofibroblast differentiation. Other key pathways included stress fiber regulation and integrin complex cell-matrix adhesion.

The multiome dataset defines cells by both expression and open chromatin. Resultantly, we sought to exploit this opportunity to identify the occurrence of unique "new peaks" (NP) in the aPT cell state and PT-S12 reference cell state, both in promoter and non-promoter regions of DEGs. A new peak was defined by the absence of open chromatin in the comparative cell state (i.e., a new peak in the aPT is a DA peak defined by fewer than 2% of cells in the PT-S12 possessing open chromatin). DA peaks occurred in 3095 regions of upregulated DEGS in PT-S12 cells and were associated with TFs identified by TRIPOD (Fig. 4f) with 380 new peaks. New peaks were only found in the PT-S12 for DEGs upregulated in the PT-S12. The converse was true in the aPT (Fig. 4g). Upregulated DEGs in the aPT contained 7535 DA peaks with an associated TF, of which 455 were new peaks not occurring in the PT-S12. The distribution of peaks and new peaks was explored across DNA methyl dip regions and histone modifications (Supplementary Fig. 10). New peak regions were more likely to occur in introns, regardless of co-occurrence with DNA methyl dips or enhancer, active, or repressed marks.

Two exemplar genes demonstrate new peak regulation (Fig. 5a, b). The two genes have 7 NP that are associated with 98 transcription factors (TFs), of which 9 are differentially expressed, as shown in Fig. 5c. In the PT-S12, *SLC5A12* (Fig. 5d) was upregulated with two new DA peaks located at 83830 (promoter) and 83824 (neighboring ANO3

gene). The 83824 peak coincided with active enhancer marks (H3K4me1 and H3K27ac) and was regulated by the TF / DEG *RREB1* and other TFs. *PROM1* was upregulated in the aPT cell type (Fig. 5e) with 5 new peaks spanning the promoter, exonic, and intronic regions. Its TFs included *PRDM1* which targeted 3 *PROM1* new peaks. *PRDM1* plays a pivotal role in regulating stem cell differentiation and tissue morphogenesis, including regulating injury signals in glomerular endothelial cells[25]. Other TFs regulating *PROM1* peaks included *ZEB1, KLF5,* and *EGR1*.

### Transcription factor networks in the proximal tubule
We sought to explore the regulatory landscape of the proximal tubule in reference to adaptive states. Multiple TFs displayed increased expression in the aPT (Fig. 6a), with concurrently increased open chromatin within a subset of these genes, such as *KLF6, ELF3, SOX4, KLF10*, and *NFKB1*. Open chromatin was calculated across each TF by including the TSS and all other peaks within 5 kb of the TSS. We conducted a TRIPOD analysis of DA peaks within the aPT. The analysis revealed ELF3 as a candidate regulatory TF of the aPT state due to its predicted regulation of multiple aPT-specific genes through key functional regulatory regions defined by DNAm and the histone landscape. Through this exploration, we discovered a predicted TF regulatory network specific to the aPT involving *ELF3, KLF6,* and *KLF10* (Fig. 6b).

For context, *ELF3* is known to regulate *Smad3*-induced fibrosis in podocytes of diabetic kidneys[26]. Both *ELF3* and *KLF6* were predicted to regulate *AQP2* expression in the collecting duct of mice[27,28]. *KLF6* has

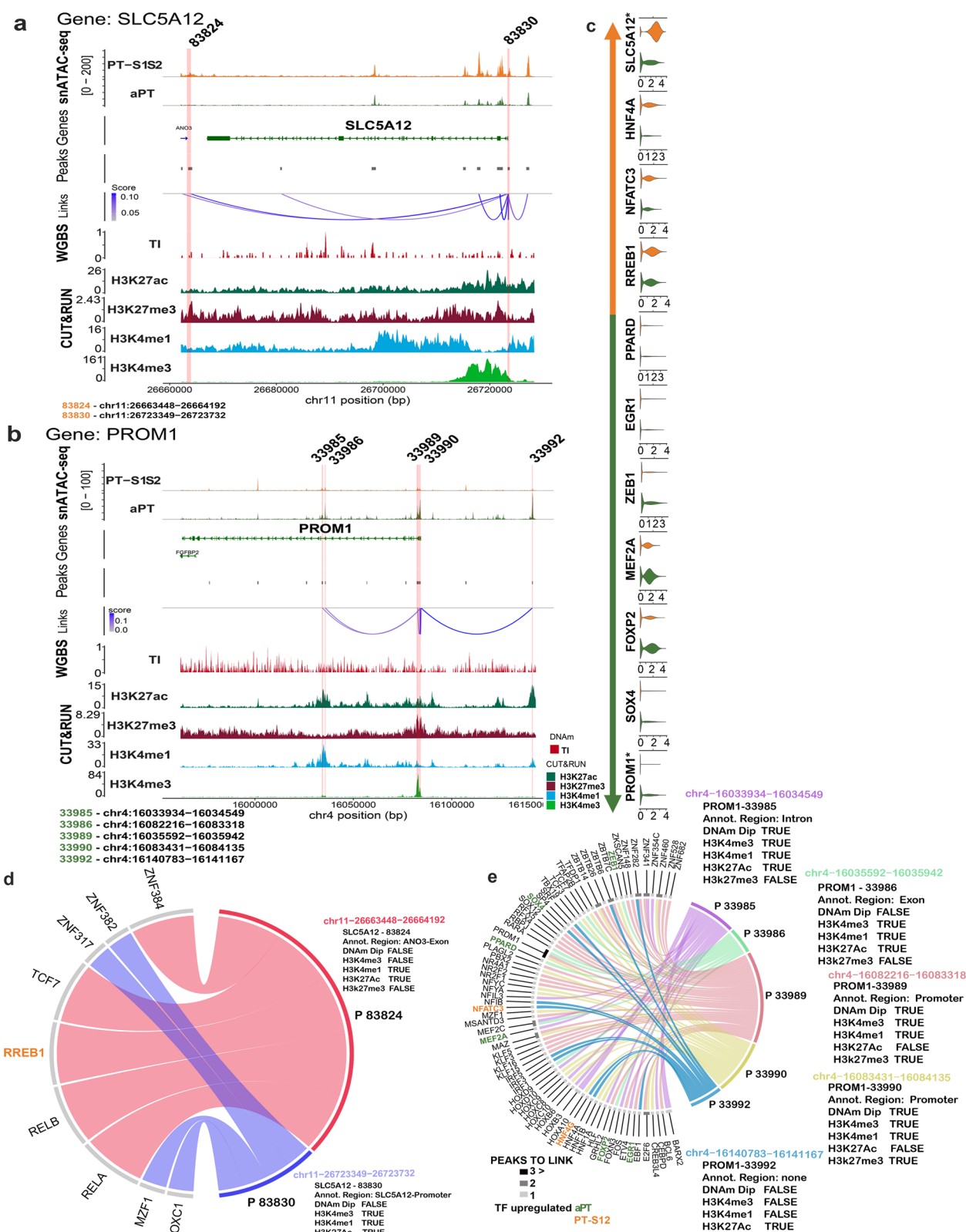

**Fig. 5 | New peaks in reference and adaptive cell states. a, b** Gene alignment of *SLC5A12*, a PT-S12 marker, and *PROM1*, an aPT marker, across snATAC-seq peaks (*N* = 12), DNAm TI dips (*N* = 15), and CUT&RUN histone marks H3K27ac (*N* = 10), H3K27me3 (*N* = 6), H3K4me1 (*N* = 3), and H3K4me3 (*N* = 3). Red stripe indicates new peaks (NP) with transcription factor (TF) binding. **c** Differentially expressed TFs of the aPT and PT-S12 which target NP of *PROM1* and *SLC5A12*. **d, e** Chord diagram of *SLC5A12* and *PROM1*, respectively, with TFs (TRUE = positive binding or DNAm Dip and FALSE = no binding or no DNAm Dip).

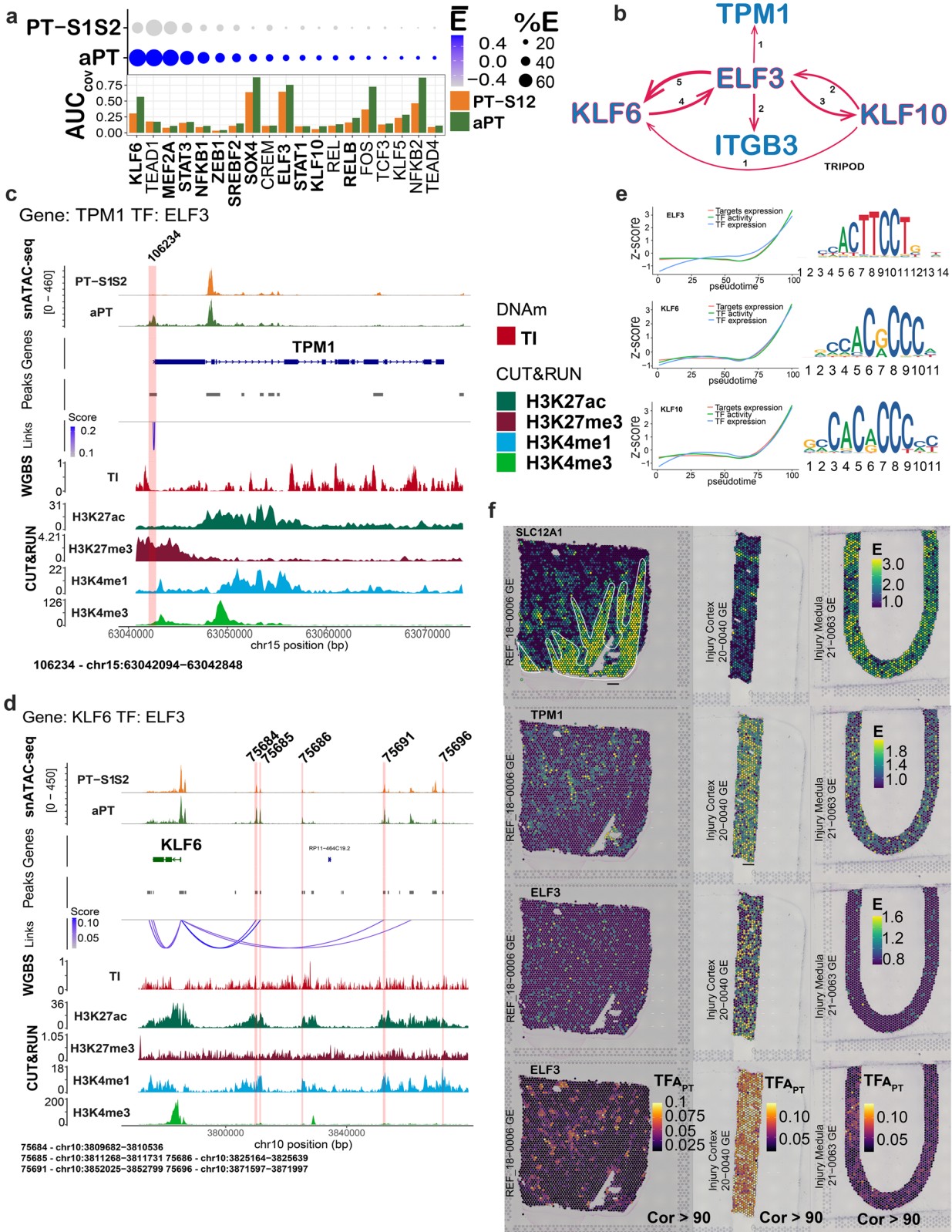

been implicated in multiple murine models of AKI, by impairing branched-chain amino acid (BCAA) catabolism in aristolochic acid I[29], by enabling pyroptosis in septic AKI[30], and in mediating apoptosis and inflammation in ischemia[31]. KLF10 has been shown to mediate epithelial to mesenchymal transition in the kidney[32] and its knockdown reduces DKD-related fibrosis[33].

*ELF3* is predicted to bind *ITGB3* at 2 snATAC-seq peaks and *TPM1* at 1 promoter-associated snATAC-seq peak, implying direct regulation of their expression. The regions *ELF3* is predicted to bind in *ITGB3* were specific active enhancer (H3K4me1+/H3K27ac+) peaks (116237 and 116240, Supplementary Fig. 10a), indicating regulation via enhancer activity rather than direct promoter activity. For *TPM1*, a new peak

**Fig. 6 | Regulation of adaptation in the proximal tubule (PT) in the multiome atlas. a** Mean expression and area under the curve (AUC) of summative open chromatin for transcription factors (TF) of the adaptive proximal tubule (aPT) and PT-S1 and S2 (PT-S12) in 12 samples. Bold font indicates expression upregulation in the aPT (negative binomial exact test, $p < 0.05$ after Bonferroni and average Fold Change >0.25 Supplementary Data 4). **b** TF network defined by the TRIPOD[1] method wherein *ELF3, KLF6,* and *KLF10* cross-regulate each other, and two genes upregulated in the aPT (*ITGB3* and *TPM1*). Edge thickness represents the number of peaks predicted in the interaction. **c, d** Alignment of epigenomic features in *TPM1* and *KLF6* for the aPT and PT-S12. Red stripe indicates a peak with predicted TF binding by *ELF3*. Co-accessibility scores were correlated with gene expression, peak accessibility by Signac, DNAm in the tubulointerstitium (TI), and histone marks. TF Peaks are numbered and correspond to (Supplementary Data 5). **e** Pseudotime trajectories from PT-S12 to aPT for the expression and activity of TFs *ELF3, KLF6,* and *KLF10* with target gene expression. *X* axis: pseudotime, *Y* axis: *z* score of transformed values based on the standard deviation of the mean. TF motifs are provided to the right. **f** Representative spatial transcriptomic mapping ($N = 3$) in a healthy reference, injured cortex, and injured medulla. *SLC12A1* defines the corticomedullary distribution (including medullary rays). *TPM1* and *ELF3* expressions are upregulated in the injured cortex. TF activity of *ELF3* is present only in the cortex (in PT dominant spots) and is upregulated in the injured cortex.

(106234) was noted just upstream of the TSS which was also a DNA methyl dip and CpG island, with histone modifications slightly offset from the center of the snATAC-seq peak (Fig. 6c). An active promoter signature aligned with the second *TPM1* peak (106235); however, the amount of open chromatin was not different between the PT-S12 and the aPT, suggesting regulation in the aPT cell state is occurring within the promoter peak 106234. Of note, the *ELF3*-regulated peaks also display H3K27me3 modification in the kidney cortex. This could be due to the bulk nature of the data or could indicate regions that are bivalently regulated for rapid transcriptional regulation switches. *ELF3* is also predicted to bind 5 snATAC peaks upstream of the *KLF6* promoter, all of which also display enhancer histone modifications (Fig. 6d) and 3 peaks outside of *KLF10* (Supplementary Fig. 10b). Analysis using scMEGA (1.0.1) predicts that both *KLF6* and *KLF10* in turn regulate *ITGB3* (Supplementary Fig. 10d, Supplementary Data 8). All 5 peaks in *KLF6* corresponded to active enhancer regulatory regions. One of the peaks in *KLF10* (67206, upstream) was a DNAm dip with repressed enhancer marks, while the others (downstream, 67188 and 67191) were active enhancers with corresponding H3K4me1 and H3K27ac peaks. Together, the TFs *ELF3, KLF6,* and *KLF10* were predicted to be involved in an aPT-specific regulatory network with cross-regulatory relationships between themselves and evidence of direct regulation of the aPT target genes *TPM1* and *ITGB3* through binding at specific promoters and enhancers (Fig. 6b).

We further explored the pseudotime and localization features of these TFs using the scMEGA package. Since it defines more permissive relationships than TRIPOD (which requires a three-way association of TF expression, target gene DE and target gene DA), we validated the method by identifying the same relationships in scMEGA with additional aPT target genes (*PROM1, VCAM1*) regulated by *ELF3, KLF6,* and *KLF10* (Supplementary Fig. 10d). Chromatin accessibility, gene expression, and TF activity are depicted in Supplementary Fig. 10d. TF activity, target gene expression, and TF expression were visualized as relationships to the pseudotime trajectory of Fig. 4b for *KLF6, KLF10,* and *ELF3* (Fig. 6e). These relationships showed strong alignment with their aPT target genes for each associated TF motif. A comprehensive TF activity matrix, filtered to retain interactions between TFs and target genes with a correlation of >90% is provided (Supplementary Data 9).

Spatial transcriptomics (ST) facilitated localization of target gene expression, TF expression, and TF activity (Fig. 6f). TF activity refers to a composite vector for a gene regulatory network (GRN) and all target gene expression, chromatin accessibility, and TF expression. We mapped these features for *ELF3* in a reference nephrectomy with cortex and medulla, as well as two diseased biopsy samples, one cortex and one medulla. Expression of *SLC12A1*, the sodium-potassium-2 chloride (NKCC) transporter in the TAL, demarcates the regions of medulla or medullary rays within the nephrectomy and differentiates the regions of the two biopsy samples. In the second row of Fig. 6f, we show the expression of *TPM1*, which is upregulated in the injured cortex, but less so in the medulla. *ELF3* expression and activity were both upregulated in the injured cortex, but not the medulla. In reference tissue, baseline *ELF3* transcription factor activity is confined to the cortex. Taken together, our results demonstrate that TF activity matrices and spatial transcriptomics are powerful tools for identifying transcriptional regulatory networks in spatially resolved tissues. *ELF3* TF activity was specifically localized to cortical regions of abundant PT-S12. These findings have important implications for understanding the molecular mechanisms underlying tissue injury and repair.

## Perturbation analysis of the proximal tubule

To determine whether the TF network of *ELF3, KLF6,* and *KLF10* is integral to or associative with the PT adaptive cell state, Celloracle[34] in silico knockout of these 3 TFs was performed individually and in combination (Fig. 7). Nine subclusters of the PT-S1, PT-S2, and aPT were identified and their connections were modeled (Fig. 7a–c). Combination knockout of *ELF3* (expression shown in Fig. 7d), *KLF6,* and *KLF10* significantly disrupted the trajectory path of both the PT-S1 and PT-S2 transition to the aPT (Fig. 7e, f). Comparing the individual knockouts of each gene, the *KLF6* knockout perturbed the largest number of aPT target genes with a similar trajectory disruption to the combined knockout (Supplementary Data 10, Supplementary Fig. 11). *ELF3* knockout also yielded a trajectory disruption between the PT-S12 and aPT, with expression changes of additional aPT target genes over the individual *KLF6* knockout. *ELF3* knockout is predicted to reduce *KLF6* expression. Thus, the combined effects are greater than either individual knockout and consistent with the cross-regulatory relationship we discovered for these TFs using TRIPOD and scMEGA. Isolated *KLF10* knockout affected the aPT trajectory to a lesser extent than either *KLF6* or *ELF3*, suggesting redundancy with these genes.

The combined knockout had minimal effect on canonical marker genes of the PT-S12, but did reduce expression of multiple aPT genes (Fig. 7g, h) with accessible chromatin and predicted binding by *ELF3* and *KLF6* (top 50 genes shown). The two aPT subclusters (aPT-A and aPT-B) demonstrated different behavior in response to the knockout. The aPT-A sub-cluster maintained higher expression of PT-S12 genes prior to knockout, but also showed less change in expression of injury genes in response to the knockout when compared to the aPT-B subcluster. In response to knockout, the aPT-B sub-cluster had increased PT-S12 gene expression (e.g., *SLC13A3, SLC34A1*) and reduced aPT gene expression. Both the aPT-A and aPT-B subclusters had reduction in aPT marker gene expression or restoration of PT-S12 gene expression after knockout; however, a greater proportion of the aPT-A cells manifest a potential repair trajectory after combined knockout. Together, these data suggest *ELF3, KLF6,* and *KLF10* mediate progression to the aPT cell state.

In a cell culture model, normal human proximal tubular kidney (NHPTK) cells were transfected with siRNA targeting the three TFs individually and in combination (Supplementary Fig. 11f). Target gene expression of *TPM1* and *VCAM1* were reduced in the combined knockout, supporting the disruption of the aPT trajectory observed in the in silico analysis. Individually, *ELF3* knockdown led to modest reductions in *KLF10, TPM1,* and *ITGB8* expression. *KLF10* knockdown led to reductions of *ELF3, TPM1,* and *VCAM1*. Individual *KLF6* knockdown led to increases in *ELF3* and *ITGB8* expression, which were not

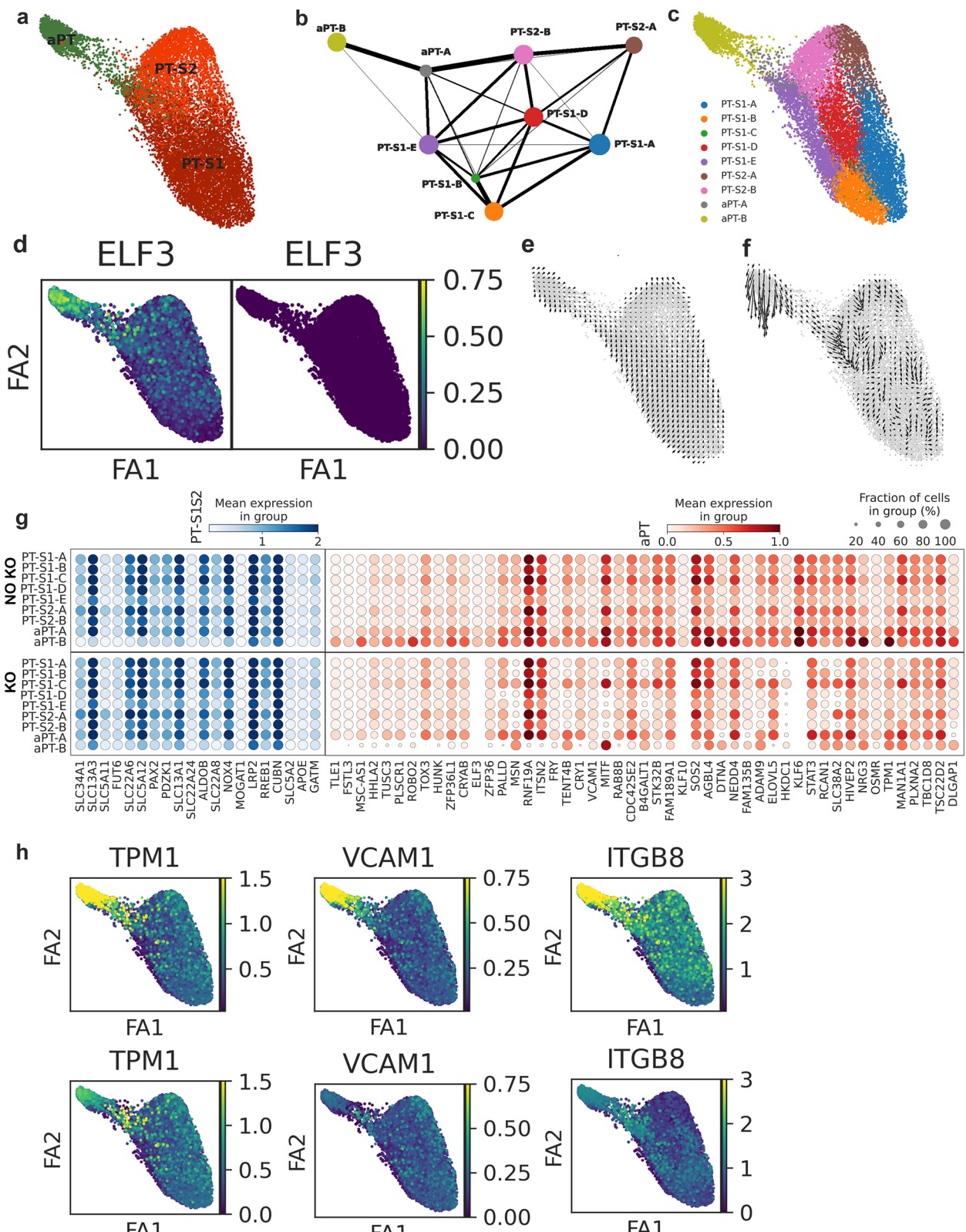

**Fig. 7 | In silico perturbation knockout in the proximal tubule (PT) (N = 12).**
**a** Cell type distribution of PT-S1, PT-S2, and adaptive PT (aPT). **b** Partition-based graph abstraction (PAGA) shows the connectivity of 9 subclusters in Louvain annotation. **c** Cell type annotation distribution across the 9 subclusters. **d** *ELF3*, a TF targeting multiple aPT marker genes, is expressed in PT clusters before combined knockout of *ELF3, KLF6,* and *KLF10,* but is reduced in expression after knockout. **e** Cell velocity combined with the pseudotime plot showing the cell flow from PT-S1 and PT-S2 to the aPT cell state. **f** The cell flow after in silico knockout revealing disruption of the trajectory. **g** Gene expression in nine subclusters of selected PT-S12 marked genes (blue) and 50 top differential aPT markers before and after combined knockout Benjamini-Hochberg adjusted *P* value < 0.05 (Wilcox test). **h** Expression of selected aPT marker genes with predicted TF binding from *ELF3, KLF6,* or *KLF10* before and after combined knockout.

seen in the combined knockdown, perhaps supporting the approach to evaluate these genes as a TF network.

## Dysregulation of the thick ascending loop of Henle in the adaptive state

In addition to the adaptive state being present in proximal tubule epithelial cells, it has also been identified in TAL epithelial cells of the Loop of Henle[3]. We sought to determine how the regulatory features of the adaptive TAL (aTAL) cell corresponded or differed from the aPT. Overall, there were fewer TAL cells than PT cells in the atlas. Nonetheless, patterns of DEGs between the cortical TAL (C-TAL) and aTAL were preserved between epithelial cells. aTAL signatures revealed loss of canonical markers like *SLC12A1*, *ESRRG*, *UMOD*, and *EGF* (Fig. 8a–c, Supplementary Data 11). We observed 3152 DEGs, with upregulation of 2485 for C-TAL and 667 for aTAL. *ELF3* was differentially expressed in the aTAL, but to a lesser extent than in the aPT. Pro-fibrotic genes were upregulated in the aTAL including *ZEB1*, *LAMC2*, and *SMAD3*. *FHL2* is known to mediate podocyte dedifferentiation and glomerular basement membrane thickening in diabetic kidney disease and is pro-fibrotic in the TI through the beta-catenin pathway[35–37], aligning with extracellular matrix-associated pathways enriched in the aTAL enrichment analysis for top 300 DEGs. Pathways showed significance with Bonferroni-adjusted *P* values < 0.05 in enrichment tests (Fig. 8d, Supplementary Data 12). *TM4SF1* was a specific aTAL DEG, but little is known of its function in the kidney. *TM4SF1* functions as a tumor suppressor[38], promotes epithelial to mesenchymal transition[39], and serves as a marker of functional human alveolar epithelial progenitor cells[40].

We explored the regulation of *FHL2* and *TM4SF1*, which shared a common TF, *NR2F1* (Fig. 8e). *FHL2* did not possess open chromatin at its TSS; instead, *NR2F1* is predicted to bind *FHL2* in an intronic region (peak 14502) before the first coding exon with an active enhancer mark. The TSS for *TM4SF1* was targeted by multiple TFs that were not differentially expressed, including *ETV1* (Supplementary Data 13). However, *NR2F1* had predicted binding upstream at peak 25256 (numbered according to Supplementary Data 14), also corresponding to an active enhancer mark. *NR2F1* was not differentially expressed (Fig. 8g), but had TF activity specific to the medulla (Fig. 8h, Supplementary Data 15). The expression of *NR2F1* in spatial transcriptomics was present in relatively few spots, corresponding to the low expression in a few cells of scRNA-seq. However, its activity was robust and localized to the medulla both in reference and injury. (Fig. 8i, lower panels), potentially highlighting the value of these complementary technologies. The aTAL phenotype included upregulation of *SMAD3*. *SMAD3* potentially interacted with *ZEB1* (at peak 85995), and while not statistically significant for DA, the peak was a DNA methyl dip and active enhancer peak. In contrast, a *SMAD3* peak targeted by *ELF3* (86016) was DA, but without a DNA methyl dip or active enhancer, suggesting alternative regulation in this region. TF expression and DA were also assessed using scMega and revealed similar patterns (Supplementary Fig. 12). Of note, *HIF1A* was associated with both the aPT and aTAL cell phenotypes. In summary, the regulation of the aTAL shares overlapping features with the aPT, but still has distinctly localized TFs that mediate the phenotype.

## Discussion

While much progress has been made in transcriptomic and proteomic interrogation of the kidney, integrating diverse epigenetic technologies is still a nascent endeavor. Multiple studies have examined single-cell open chromatin in the context of mRNA expression[15,41–44]. Open chromatin is necessary for gene transcription, but it may also be found at promoters or enhancers of inactive or silent genes, indicating that chromatin accessibility is insufficient alone to predict expression[6]. There are multiple layers of epigenetic regulation that determine the activity of regulatory regions, with DNA methylation and histone

modifications being among the most important[45]. Relevant studies by the Susztak laboratory have examined WGBS with chromatin immunoprecipitation sequencing of histone modifications in diabetic and healthy tissue samples[10] as well as interpreted genome-wide association study variants in the context of snATAC-seq and methylation. We build upon this investigation by integrating CUT&RUN, WGBS, multiomic snATAC-seq, regional transcriptomics, spatial transcriptomics, and regional proteomics, all at a biopsy scale of tissue. In this study, we have generated a genome-wide dataset that integrates all these epigenomic layers of the gene regulatory program with mRNA and protein expression in overlapping human kidney samples.

A major goal of this work was to create a comprehensive atlas that will serve as an essential reference for comparison to clinical biopsy samples and diseased kidney tissue. Studies of this type are needed to reveal how cell-type specific regulatory regions control the transition between healthy and diseased states, and to guide efforts to reprogram cells to promote repair after injury. Because our work builds upon a transcriptomic atlas of healthy and diseased tissue[3], we were able to explore the regulatory network of the adaptive cell state that is associated with the progression of CKD and AKI to CKD transition. Although this epigenomic atlas includes a limited number of diseased kidney biopsy samples, both the reference and adaptive cell states are found in nephrectomy samples, which have varying degrees of injury. Healthy tissue is heterogenous and often contains abnormal pathology and altered cells with injury signatures. Most samples selected in this study were provided by the KPMP and HuBMAP, supplemented by local repositories, in order to assess the technical reproducibility and potential biological insights of these technologies to each consortium. Future studies in a larger KPMP cohort of diseased tissue will better examine specific epigenomic alterations in AKI, CKD, and those associated with histologic manifestations. For example, DNA methylation is known to correlate with renal disease and outcomes, including those observed during ischemia in kidney transplant allografts[46].

Our bioinformatics analyses highlight the potential value of our dataset to elucidate regulatory pathways involved in the adaptive cell state. We uncovered differential regulation of the PT and aPT, specifically identifying a TF network of *ELF3*, *KLF6*, and *KLF10*, responsible for multiple downstream target genes within the adaptive cell state and localizing specifically to the kidney cortex and PT. In silico knockout of the TFs in this network supported cross-regulation among these genes and revealed two distinct aPT trajectories. Interestingly, target gene expression in one of these trajectories was significantly more responsive to TF knockdown, underscoring a potential role in transition between PT and aPT. Further, our studies revealed that regulation of the adaptive cell state differs across epithelial cell types, as the TF regulation of the aPT was different than that of the aTAL. The multiome defines cells based on mRNA signatures and TSS open chromatin, unlike standard snATAC-seq, which uses the TSS alone to define cell types. This added dimension allows for greater interpretation of peaks outside the TSS and improves resolution of injury cell states in which open chromatin of canonical marker genes may be lost. The integration of WGBS and CUT&RUN facilitates the assessment of DNA methylation and histone modifications as contributions to chromatin accessibility.

In order for the atlas to serve as a foundation for the scientific community, we reasoned that it must meet specific benchmarks. Until recently, epigenomic analysis of clinical tissues has been limited by the relatively large amount of starting material needed to perform some of the assays. However, advances in epigenomic technologies have now enabled these assays to be performed on a smaller scale. We performed all methods described in this study using relatively low cell inputs suitable for interrogation of clinical kidney biopsy samples. For example, all nephrectomy tissue was cut to be biopsy-sized and in a subset of samples, multiple technologies were performed on a single

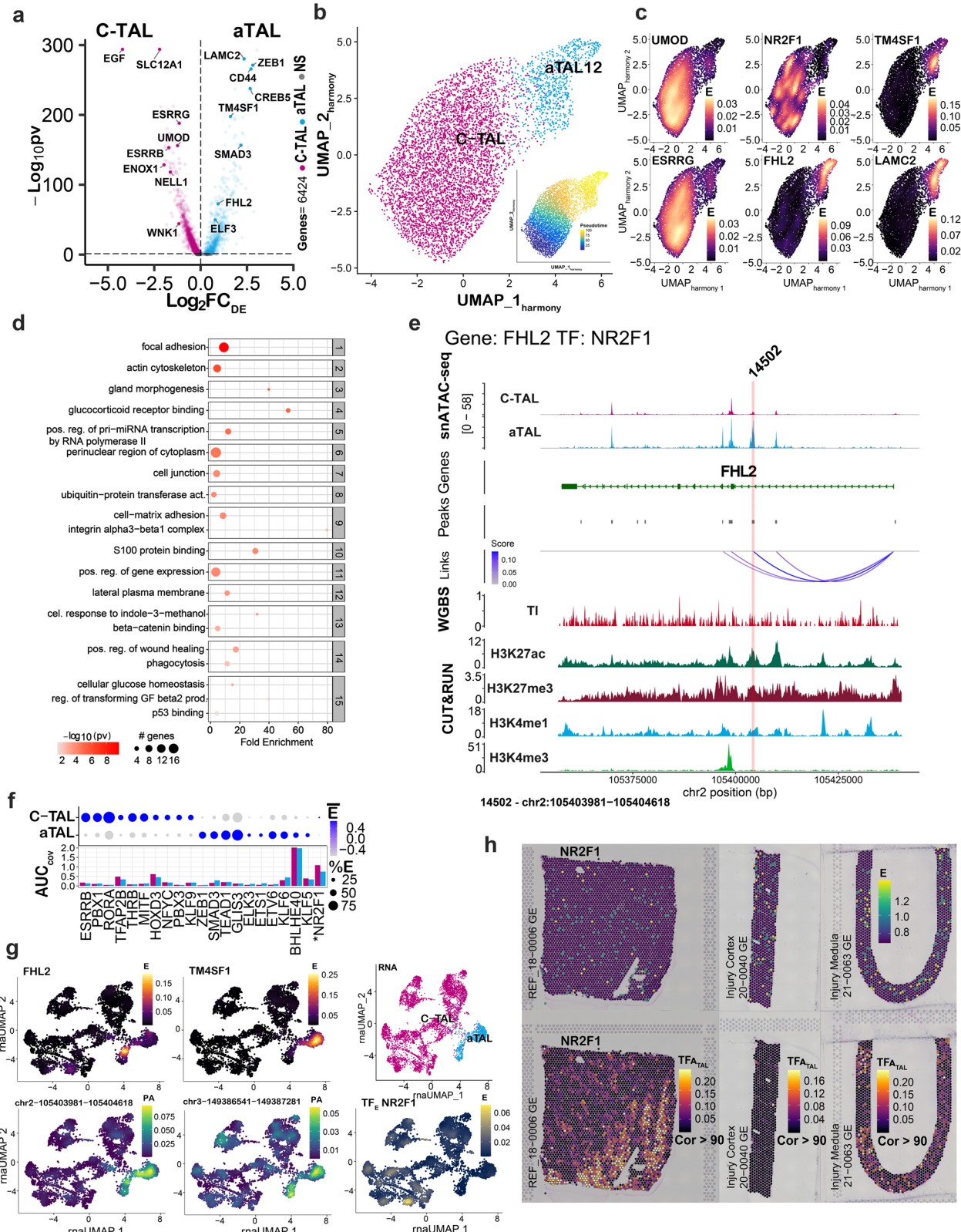

Optimal-Cutting-Temperature compound embedded block. A second key feature of a reference dataset is that the methodology must meet rigorous QA/QC criteria to account for assay drift and batch effects, and be readily reproducible by other investigators. Each of the technologies used in this study has undergone careful expert review and standardization based on published KPMP criteria[21] and the resultant quality data is included in the supplement. In this study, we

demonstrated a high degree of reproducibility among technical and biological replicates collected over or stored for many months. Furthermore, we demonstrated genome-wide orthogonal validation between complementary epigenomic features. Investigators using our opensource detailed protocols[47-49] adapted to low cell numbers will be able to validate our data in independent reference samples and use these methods to interrogate additional diseased kidney tissue.

**Fig. 8 | Adaptation in the cortical thick ascending loop of Henle (C-TAL).**
**a** Differentially expressed genes (DEGs) between the adaptive TAL (aTAL, blue) and C-TAL (magenta), for N = 3152 genes at a Bonferroni adjusted P value < 0.05 (Wilcox test) within the multiome atlas (N = 12). **b** UMAP harmony of aTAL and C-TAL. Inset shows pseudotime from C-TAL to aTAL. **c** Gene expression localizes in aTAL cells for aTAL marker genes (*LAMC2*, *FHL2*, and *TM4SF1*). Canonical C-TAL markers (*UMOD* and *ESRRG*) are expressed in the TAL. *NR2F1* is a TF that is not differentially expressed. **d** Top 15 clusters from GO-All pathway enrichment analysis at a Bonferroni adjusted P value < 0.05 (enrichment tests). The genes are based on DA regions in the aTAL. Key pathways of the adaptive process overlap with those of the aPT, including mesodermal cell differentiation and adhesion. **e** Alignment of epigenomic features in *TM4SF1* and *FHL2* for the aTAL and C-TAL.

The red stripe indicates a peak with TF binding by *NR2F1*. Co-accessibility scores were correlated with gene expression, peak accessibility by Signac, DNAm in the tubulointerstitium (TI), and histone marks. Additional TF peaks and target genes are found in Supplementary Fig. 12. TF Peaks are numbered and correspond to Supplementary Data 13. **f** Top 10 most differentially expressed TF in the aTAL and C-TAL (by TRIPOD1). The TF *NR2F1* is not differentially expressed. The bar plot conveys the AUC of summative gene open chromatin. **g** The expression of *FHL2* and *TM4SF1* is highest in the aTAL. The peak activity of the peaks targeted by *NR2F1* is also highest in the aTAL region. **h** Representative spatial transcriptomic mapping (N = 3) in a healthy reference, injured cortex, and injured medulla. *NR2F1* expression and activity occur in the medulla, and activity is present in injury.

This multimodal analysis of chromatin regulation aligned histone modifications with DNA methylation patterns and open chromatin across the genome of the kidney. There is not a one-to-one correlation of these technologies and their integration provided a holistic view of the contributors to chromatin accessibility. This comprehensive epigenetic map will complement the interpretation of signatures within the merged Human Biomolecular Atlas project (HuBMAP) and Kidney Precision Medicine Project (KPMP) transcriptomic atlas[3]. Future studies comparing healthy and diseased tissue have the potential for significant clinical impact by uncovering epigenetic biomarkers of disease progression and druggable cell-specific regulatory targets that promote successful or failed repair.

## Methods

### Human subjects and samples

This study complies with all relevant ethical regulations and was approved by the Institutional Review Boards of Indiana University and Washington University in St. Louis and as part of sIRB protocols of the HuBMAP and KPMP. Human samples and clinical data were derived from the following sources: (1) The Biopsy Biobank Cohort of Indiana (BBCI, IRB #1906572234)[50] approved by Indiana University, (2) the Kidney Translational Research Center (KTRC, (IRB #201102312) of Washington University in St. Louis[51], (3) the Kidney Precision Medicine Project (KPMP)[52], and (4) the Human Biomolecular Atlas Project[53] (HuBMAP). For the BBCI, KTRC, and HuBMAP, samples were obtained under a waiver of informed consent due to minimal risk to subjects according to the United States Department of Health and Human Services Common Rule 45 CFR 46.116(f). For KPMP, samples were acquired with written informed consent. Samples included healthy reference tissue from patients undergoing total nephrectomy, deceased donor nephrectomy, percutaneous kidney biopsy in a healthy transplant donor, or percutaneous kidney biopsy in an individual with kidney disease (Supplementary Data 1). All nephrectomy samples were dissected from tumor-free regions. Individuals with AKI or CKD were included in the KPMP. The proportion of cortex and medulla, the demographic data, the clinical data, and the histopathologic findings were recorded.

### Multiome sample processing, library preparation, sequencing, and analysis

For sample processing, in 12 total samples (8 distinct biologic samples), 47,217 nuclei were isolated from tissue cryosections according to a published protocol[54], and with the following exceptions: (1) tissue sections were cut then stored on dry ice until nuclei isolation; (2) Protector RNase Inhibitor (Sigma-Aldrich, Catalog #3335402001) was used at a concentration of 1.0 U/µl; (3) Complete protease inhibitor cocktail (Roche, cat #11836153001) was included for final 1× concentration; and (4) 4′,6-diamidino-2-phenylindole (DAPI) was excluded from the extraction buffer. Furthermore, two nuclei samples were further purified immediately following isolation using the LeviCell system (Levitas Bio) using company protocols. This was performed to assess for any data quality improvements for nuclei that had been

separated from cellular debris generated during the tissue dissociation.

For library preparation, isolated nuclei were processed immediately using the Chromium Next GEM Single Cell Multiome ATAC + Gene Expression (v.1.0) kit according to the step-by-step protocol available online[49].

For sequencing, the RNA and ATAC libraries were sequenced separately on the NovaSeq 6000 (Illumina) system (NovaSeq Control Software v.1.7.0 and v.1.7.5). Sample demultiplexing, barcode processing, gene expression and open chromatin peak quantifications were performed using the Cell Ranger Arc software (v.2.0.0) using GRCh38 (hg38) reference genome.

For RNA Analysis, cell barcodes passing the following quality control filters were used for downstream analyses: (1) passing 10X Cell Ranger Arc (RNA/ATAC) filters; (2) considered singlets by the DoubletDetection software (v.2.4.0); 3) showing greater than 400 and <7500 non-mitochondrial genes detected; (4) passing a gene UMI ratio filter (gene vs. molecule.cell.filter) from the Pagoda2 software (https://github.com/hms-dbmi/pagoda2). Mitochondrial transcripts were removed, and the query RNA counts were integrated with a previous snRNA-seq reference atlas of healthy and injured cell types in the human kidney[3]. This integration was performed using Seurat (v.4.0.0)[3]. Briefly, RNA counts were normalized using sctransform, anchors between datasets were identified based on the reference Pagoda2 principal components. Query data was then projected onto the reference principal component (PC) and UMAP space, with cell type labels transferred using the MapQuery function. A k-nearest neighbor graph (k = 100) was generated using Pagoda2 based on the integrated PCs, and integrated clusters were identified using the infomap community detection algorithm. The query portion of each cluster was then annotated to the most overlapping, correlated and / or predicted reference cell type label. This was performed to account for inaccurate cell type labeling that is prevalent for altered cellular states. A further step of manual assessment of cell type markers was performed to confirm identities. Integrated clusters that showed markers or identities that overlapped across disparate cell types were labeled as ambiguous or low-quality and removed.

For chromatin Analysis, snATAC data was processed using Signac (v.1.6.0)[55]. Peaks called using Cell Ranger Arc were combined across experiments using the reduce function. Fragment objects for each experiment were prepared from Cell Ranger Arc fragment files using the CreateFragmentObject function. The combined set of peak regions was used to generate peak-by-cell matrices for each experiment using the FeatureMatrix function. These peak matrices were then used to create individual Seurat objects that were merged to form a single combined object. Only cell barcodes that were retained from RNA analyses were used for further analyses. Accessible peaks were then called separately for multiple levels of cell type annotations (clusters, subclass level 3 and subclass level 1) using the CallPeaks function and MACS (v.3.0.0a6; https://github.com/macs3-project/MACS). All peaks, including those called by Cell Ranger Arc, were combined using the reduce function and filtered to remove: (1) regions >10000 and <20

base pairs; (2) regions falling within nonstandard chromosomes; (3) regions occurring in blacklist regions using the blacklist_hg38_unified object from Signac. The final peak set was used to create a new peak by cell count matrix and seurat object as detailed above. Gene annotation of the peaks was performed using GetGRangesFromEnsDb(ensdb = EnsDb.Hsapiens.v86). Nucleosome signal (NS) scores, transcription start site (TSS) enrichment scores and fraction of reads in peaks (FRiP) were calculated for each cell using the NuceosomeSignal, TSS Enrichment and FRiP functions. Cell barcodes further passing the following ATAC filters were kept for downstream analyses: (1) >1000 and <100000 peak counts per cell; (2) a FRiP greater than 0.25; (3) an NS score less than 4; (4) a TSS enrichment score >2. A combined Seurat object was then generated with separate RNA and ATAC assays for the same cell barcodes. The LinkPeaks function was used to link potential regulatory peaks to each gene, taking into account GC content (computed using the RegionStats function with the BSgenome.Hsapiens.UCSC.hg38 genome), overall accessibility and peak size. Transcription factor motif information was added to the Seurat object using the AddMotifs function in Signac and using Jaspar motifs (JASPAR2020, all vertebrate). Motif activity scores were computed using chromVar[56] (v.1.12.0; https://greenleaflab.github.io/chromVAR) through the RunChromVar function. For quality control assessment, the multiome dataset was compared to a publicly available snATAC-seq dataset (GSE151302) that was aligned and merged with the KPMP snRNA-seq atlas[3].

## Cleavage under targets & release using nuclease (cut&run) sample processing, library preparation, sequencing, and analysis

For sample processing, liquid nitrogen or OCT-embedded frozen kidney tissue was processed according to a modified protocol adapted from Epicypher and available on protocols.io[48]. Antibodies used in this study for CUT&RUN reactions: H3K27ac (Cell Signaling, 8173), H3K27me3 (Cell Signaling, 9733), H3K4me1 (Cell Signaling, 5326), H3K4me3 (Cell Signaling, 9751), and IgG (Cell Signaling, 2729) at a 1:50 dilution. The number of samples processed for each antibody are: H3K27ac $N = 6$ total and 4 distinct biologic, H3K27me3 $N = 10$ total and 6 biologic, H3K4me1 $N = 3$ total and 1 biologic, H3K4me3 $N = 3$ total and 1 biologic.

For library preparation, up to 3 ng of DNA from KPMP donor samples was used to prepare sequencing libraries using the Ion Plus Fragment Library Kit (Thermo Fisher Scientific, 4471252), following manufacturer's instructions, using 17 cycles to amplify the library. Up to 0.5 ng of DNA from KTRC donor samples (3399, 3409, 3447) was used to prepare sequencing libraries using NEBNext Ultra II Library Kit (NEB, E7645), following manufacturer's instructions, using 18 cycles to amplify the library.

For sequencing, 100 pM KPMP donor libraries were sequenced on the Ion Torrent Proton to a sequencing depth of 30 million reads (2–3× genome coverage). 0.8 pm KTRC donor libraries were sequenced on the NovaSeq 6000 (Illumina) targeting 100 million paired-end reads.

For analysis, detailed bioinformatic analysis with command line examples for Illumina sequences can be found on protocols.io[48]. Briefly, trimmed fastq files were aligned to the hg38 reference genome using Bowtie2 (for Illumina sequenced) or TMAP (for Ion Torrent sequenced). Aligned reads were extracted and converted to a sorted bam file using SAMtools. Scaled bigWig files were generated using Deeptools, and peaks were called using Macs2.

## Whole genome bisulfite sequencing (Wgbs) sample processing, library preparation, sequencing, and analysis

For sample processing, thirty kidney glomerular and tubulointerstitial (TI) samples from 14 distinct donors were processed along with positive and negative blood controls. Glomerular and TI samples from 1 donor were processed twice to assess batch effect. Tissue was stored at −80 °C and sectioned from an OCT block at 12 μm thickness, and mounted on Leica PPS-membrane Laser Microdissection (LMD) slides (Leica, Cat# 11505268). A rapid stain protocol[23] prepared slides for LMD, briefly: slides were fixed in −20 °C Acetone (Sigma-Aldrich, Cat# 270725-1 L) for 1 minute, washed with PBS (VWR, Cat# K812-500ML) 2 times for 30 seconds, stained with antibody mix [4 μL OG-Phalloidin (Oregon Green 488, ThermoFisher, Cat# O7466) + 1.5 μL DAPI (ThermoFisher, Cat# 62248) in a total volume of 200 μl of PBS solution] for 5 minutes, washed with PBS for 30 seconds twice and air dried for 5 minutes. Glomerular (GLOM) and TI sections were dissected on a Leica LMD6500 system and collected directly in the flat cap of an autoclaved 0.5 mL microcentrifuge tube (ThermoScientific, Cat# AB-0350) tubes, then stored in −80 °C. DNA was isolated using the Pure-LinkTM Pro 96 Genomic DNA Kit (Cat# K1821-04A) with minor modifications. Tissue lysates were prepared using the "Mammalian Cells and Blood Lysate" protocol, briefly: 200 μl of PBS, 20 μl of Proteinase, 20 μl of RNase A were added to each tube, mixed and incubated for 2 minutes at room temperature. Then, 200 μl of PureLinkTM Pro 96 Genomic Lysis/Binding Buffer was added to each tube, mixed and incubated for 15 minutes at 55 °C. The produced tissue lysate was mixed with 200 μl of 100% ethanol and then transferred to a Pure-LinkTM gDNA filter plate, centrifuged at ≥2100 × $g$ for 10 minutes, then washed with 500 μl of wash buffer 1 and wash buffer 2 separately; the purified DNA was eluted in 50 μl of elution buffer.

For library preparation, DNA quantity and quality were assessed by Qubit Fluorometer 4.1 (Invitrogen). Bisulfite conversion was performed using the EpiTect Fast DNA Bisulfite Kit (Qiagen Cat# 59824). An input of 10 ng was used for library preparation, and QIAseq Methyl DNA Library Kit (Cat# 180502) was used for amplification and ligation of Illumina adaptors. Library purification was performed using QIAseq Beads. The library quality was assessed using an Agilent Bioanalyzer 2100 and Qubit Fluorometer 4.1.

For sequencing, multiple libraries were pooled in equal molarity, and a final concentration of 300 pM was loaded onto the NovaSeq 6000 sequencer (Illumina) for 150 bp paired-end sequencing. Approximately 200 million read pairs (400 million reads total) were generated per library for ~15–20× calculated coverage. The actual final measured coverage averages 11.8× after processing. An EpiTect Unmethylated DNA control (Qiagen Cat# 59568) and EpiTect Methylated DNA control (Qiagen Cat# 59655) were sequenced as controls.

For analysis, sequencing quality control, mapping, and methylation analysis were performed using the pipeline with FastC (v.0.11.5) and MultiQC (v.1.8). TrimGalore (v.0.6.7) was used to remove Illumina adapter sequences and ten 5′ prime bp and five 3′ prime bp. Bismark (v.0.23.1) DNA non-directional mapping was used to align to the hg38/GRCh38 reference genome. Duplicated reads were removed. Methylation extractor was used to define CpG regions. Alignment was performed on the Carbonate Cluster at Indiana University. Batch effect correction was performed by an empirical Bayes framework[57] by SVA R package. To test reproducibly between batches, biological replicants were applied to the Rank−Rank Hypergeometric Overlap (RRHO2)[58] to determine agreement.

DNA methylation data was stored in a methylKit (v.1.20.0) object. This object accepted CpG sites with coverage greater than 5 reads across all samples for the GLOM and TI. Whole genome DNAm was extracted (Supplementary Fig. 1) for each nucleotide. Methylated cytosines ($\#C's$), unmethylated cytosines ($\#T's$) and coverage ($\#C's + \#T's$) were quantified per nucleotide in the glomerulus ($g$) and tubulointerstitium ($ti$) for each sample $i$. Regions of interest were extracted using Grange GenomicRanges (v.1.46.1), where each row has information for genomic regions as small as one base pair to those spanning an entire chromosome. Regions were subset and summed (coverage, numC's, and numT's). For each region, the methylation

levels are given by Eq. (1) GLOM and Eq. (2) in TI:

$$ml_{GLOM} = \frac{\sum_{i=1}^{15} \#C's\,g_i}{\sum_{i=1}^{15} \#T's\,g_i + \#C's\,g_i} \tag{1}$$

$$ml_{TI} = \frac{\sum_{i=1}^{15} \#C's\,ti_i}{\sum_{i=1}^{15} \#T's\,ti_i + \#C's\,ti_i} \tag{2}$$

A $t$ test is applied with multiple testing correction for $ml_{GLOM_i}$ and $ml_{TI_i}$.

The relative, differential, or hyper-methylation (*Hyper*) is given by Eq. (3):

$$Hyper = \log_2\left(\frac{ml_{GLOM} + \beta}{ml_{TI} + \beta}\right) \tag{3}$$

were $\beta = 0.001$. The measurement compares the GLOM and TI compartments to assess relative methylation levels. A positive value points to greater methylation in the GLOM and a negative value has greater methylation in the TI. *Hyper* was calculated for any region.

Using the annotatr (v.1.21.0) R package (PMID: 28369316), genes were annotated with whole gene and gene regions (1 to 5 kb upstream of TSS, exons, introns, promoters, 3′ UTRs, 5′ UTRs, CpG islands, CpG shelves, CpG shores, CpG intergenic). Methylation status was summed across CpGs in a region for each gene.

### Regional transcriptomics sample processing, library preparation, sequencing, and analysis

For sample processing, an overlapping set of samples (11 of 14 distinct biologic WGBS samples) were processed for regional transcriptomics of GLOM and TI according to a laser microdissection protocol[23], available on protocols.io[47]. Briefly, glomeruli and the tubulointerstitium were microdissected from a frozen OCT section using a rapid staining antibody-based protocol.

For library preparation, cDNA was synthesized with the SMARTer Universal Low Input RNA Kit protocol (Clontech, cat. no. 634938) using the standard RiboGone–Mammalian Kit protocol.

For sequencing, barcoded libraries were pooled in equal molarity and sequenced on an Illumina HiSeq 4000 for about 100 million reads per library.

For analysis, mRNA expression was calculated according to a negative binomial dispersion exact test. Statistics were performed with negative binomial generalized linear models[59] in edgeR (v.3.36.0).

### Regional proteomics sample processing, protein measurement, and analysis

For sample processing, an overlapping set of samples (8 of 14 distinct biologic WGBS samples) were processed for regional proteomics of GLOM and TI according to the protocol available on protocols.io (https://doi.org/10.17504/protocols.io.bew6jfhe). Briefly, glomeruli and the tubulointerstitium were microdissected from a frozen OCT section. Protein content was measured by Liquid chromatography-tandem mass spectrometry. Analysis was performed with an Easy-nLC 1000 coupled to an Orbitrap Fusion mass spectrometer (Thermo-Scientific, Waltham, MA).

For protein measurement and analysis, protein was quantitated by high-performance liquid chromatography with mass spectrometry[10].

### Spatial transcriptomics sample processing, library preparation, sequencing, and analysis

For sample processing, human kidney tissue ($N = 3$) was prepared and imaged using the Visium Spatial Gene Expression protocols (10x Genomics, CG000240 protocol)[60]. From optimal-cutting temperature

(OCT) compound embedded blocks, tissue was sectioned at a thickness of 10 μm. Hematoxylin and eosin (H&E) stained brightfield Images were acquired from a Keyence BZ-X810 microscope. The microscope was equipped with a Nikon 10X CFI Plan Fluor objective. Brightfield mosaics were stitched and aligned with Visium fiducials. mRNA was isolated from the tissue sections after 12 minutes of permeabilization. mRNA was bound to oligonucleotides in the fiducial spots and then reverse transcribed.

For library preparation and sequencing, the mRNA underwent second-strand cDNA synthesis, denaturation, cDNA amplification, and SPRIselect cDNA cleanup (Visium CG000239 protocol). cDNA was sequenced on an Illumina NovaSeq 6000.

**Analysis.** Space Ranger (v2.0.0) with the reference genome GRCh38 2020-A was used to perform expression anEalysis, mapping, counting, and clustering. To localize transcription factor activity, we first created trajectories to obtain gene regulatory networks (GRNs) in scMEGA[61]. The multiome data was subdivided into aPT/PT-S1S2 cells and aTAL / C-TAL cells to identify the pseudotime spectrum of cells from health to adaptive injury states. Harmony was applied for batch correction[62]. The R package Destiny[63] was used to construct a diffusion map and apply a reduction in non-linear space. Computational inference of GRNs was obtained in order to map TF activity with scMEGA. Briefly, a composite vector accounting for target gene expression, target gene open chromatin accessibility, and TF expression are localized over renal tissue using spatial transcriptomic expression patterns.

### Cohen's Kappa measurement of alignment

All datasets were aligned to the hg38/GRCh38 reference genome. The agreement of multiome peaks, each CUT&RUN antibody peak, and WGBS dips were assessed. Regions were defined as peak or no-peak in the multiome dataset by Signac. To evaluate peak alignment, the podocyte (POD), proximal tubule (PT-s1/S2), and cortical TAL (cTAL) cell types were defined with the findmarker function and cell-type specific marker genes were selected from the merged HuBMAP/KPMP sc/snRNAseq atlas v 1.0[3]. A Gisch filter was implemented to select cell-type specific peaks for the multiome and remove background signals from other technologies. For this filter, the AUC was calculated for each gene region (exon, intron, etc.). In regions where the peak AUC was greater than the average AUC of the entire gene, the region was called a peak. Other regions were considered no-peak regions. Peaks were called similarly for H3K27ac, H3K27me3, H3K4me1, and H3K4me3 antibody marks. WGBS dips were called when the AUC of a region was less than the average AUC of the whole gene. Fisher's exact and Cohen's Kappa tests were used to evaluate the agreement of peaks across technologies.

### Heatmap analysis

Heatmaps of the alignment of histone modifications with genomic features were generated using DeepTools. Genomic features assayed were restricted to those that overlapped with a macs2 called ATAC peak (bulk ATAC-seq or PT-specific snATAC-seq peaks). Promoters were defined as a 2 kb region centered on an annotated GRCh38 transcription start site. Putative enhancers were called by the overlap of macs2 called H3K4me1+/H3K27ac+/H3K4me3- bulk ATAC-seq peaks not located within an annotated gene body or promoter. DNA methylation dips were called based on a DNAm value of ≤0.4 across a 50 bp region or larger.

### Cell-type deconvolution

Cell-type deconvolution[6] was performed using the Dtangle algorithm in the Granulator (v.1.7.0) R Package. The RNA Signature was derived from the merged HuBMAP/KPMP snRNAseq kidney atlas. The top 10% marker DEGs were selected for which CUT&RUN promoter peaks were detected in H3K27ac, H3K4me1, or H3K4me3. Cell types were

restricted to those with ≥20 Marker Genes in this top 10% of DEGs. The descending thin limb and endothelial cell clusters were removed because of similarity to TAL signature and poor algorithm performance, respectively. The Detangle algorithm then identified the percentage of cells from each cell cluster based on the histone mark.

## Transcription factor analysis

TRIPOD (v.0.01) R package was employed to identify regulatory trios from the multiome dataset between: (1) open chromatin with cis-regulatory regions of target genes (100–200 kb surrounding the TSS), (2) target gene expression, and (3) TF expression[24]. The TF definitions were based on the Signac enrichment method[55] and the JASPAR2020 dataset[64]. TRIPOD creates a meta cells matrix to link RNA expression (snRNAseq) and open chromatin (snATAC-seq). Briefly, the steps in the TRIPOD method include: (1) RNA is normalized by regularized negative binomial regression, nuclei are clustered by PCA and UMAP, (3) the open chromatin in the snATAC-seq assay is normalized by default options, and (4) using the PCA (RNA) and LSI (ATAC) reduction, a weighted nearest neighbor (WNN) graph was built[65] to cluster RNA and ATAC together. The matrix is used to compare each regulatory trio and identify TF and target gene associations with high confidence.

For pseudotime trajectories of transcription factor activity and expression, scMEGA (v.1.0.1) was used as described above[61]. For differentially expressed genes, a Seurat findmarker function was run using the test.use = wilcox and plotted as a volcano plot with a Bonferroni-corrected $P$ value < 0.05 and percent of cells expressed above 2%.

To calculate the differential accessibility (DA) of genes, the Seurat function FindMarkers was used with the logistic regression LR. Peaks were classified as DA peaks with a Bonferroni-corrected $P$ value < 0.05 and an absolute $\log_2$ fold change >0 between the cell types, with the setting only.pos = FALSE. An average $\log_2$ fold change >0 means greater open chromatin in adaptive injury cell states.

## In silico perturbation

Co-accessibility networks for the PT cells were obtained with Cicero[66]. Following the CellOracle (v.0.10.12) pipeline[34], 9 subclusters were identified, and the connections in gene expression were modeled with a graph abstraction. A combined knockout of *ELF3*, *KLF6* and *KLF10* was simulated in silico, and changes in gene expression and cell velocity were calculated. Highly aPT-changing genes were selected by the higher variation in Fold Change between aPT and PT-S12.

## Expression modeling

DNA methylation was summed across CpG sites within gene region annotations for whole gene bodies ($N = 30,024$), exons (27,830), introns (25,904), promoters (28,625) and CpG Islands (18,849). Methylation was compared to mRNA and protein expression derived from a set of completely overlapping samples in differentially expressed genes (DEGs, $N = 5942$) or proteins ($N = 1917$) between GLOM and TI dissections. The correlation of differential DNA methylation with expression of DEGs or DEPs was assessed by univariate linear regression, and residuals were calculated using ggplot2 and stat R packages. A best-fit regression model was developed wherein a gene annotation (CpG island, exon, intron, promoter) was selected based on the greatest C-statistic between $\log_2$ differential expression and differential gene region methylation for each gene (Fig. 2c, Supplementary Fig. 5).

Analogously, DNA methylation was summed within four sets of CUT&RUN peaks, bulk ATAC-seq peaks, and single-cell multiome peaks (PT-S1 and C-TAL for TI, and Podocyte for GLOM). The peaks of these orthogonal technologies acted as filters for univariate regressions in which summative DNA methylation was plotted within each peak set against mRNA or protein expression for all DEGs and DEPs.

## Enrichment analyses

pathfindR (v.2.1.0) was used to combine a BioGRID active subnetwork search with a gene ontology (GO) pathway enrichment analysis. We input a gene interest list with adjusted $P$ value from gene differential expression[67]. The enriched terms are clustered and identify significative adjusted $P$ value < 0.05 and are called representative pathways and their members. The $P$ values obtained from the enrichment tests are adjusted by Bonferroni method.

## Kidney cell culture

Normal human proximal tubular kidney (NHPTK) cells[68] were plated at a density of $4.5 \times 10^4$ in a 24-well flat-bottom plate and maintained in Renal Epithelial Growth media (REGM, Lonza, Basel, Switzerland) and 9% fetal bovine serum (HyClone). Cells were diluted to 20–30% confluency three times a week and maintained at 37 °C in 95% humidified atmosphere with 5% $CO_2$.

## siRNA knockdown and real-time polymerase chain reaction

We conducted siRNA-mediated knockdown of *ELF3*, *KLF10*, and *KLF6* in NHPTK cells with lipofectamine. Cells were plated on Day 1 with lipofectamine and a pool of up to 6 directed siRNA constructs (20 nM concentration each) for 5 conditions: (1) Silencer, (Cat. No. 4390843) scrambled siRNA control, (Cat. No. 4427037), (2) a pool of two *ELF3* siRNA molecules (siRNA ID s4623 and s4624), (3) a pool of two *KLF6* siRNA molecules (siRNA ID s3376 and s3375), (4) a pool of two *KLF10* siRNA molecules (siRNA ID s14129 and s14130), and (5) a combination of 6 siRNA molecules targeting *ELF3, KLF10*, and *KLF6*. Gene expression of (Cat. No. 4448892) *ELF3* (Assay ID Hs00963877_g1), *TPM1* (Assay ID Hs04398572_m1)*, ITGB8* (Assay ID Hs00174456_m1), *(*Cat. No. 4453320)*, KLF10* (Assay ID Hs00921811_m1), *KLF6* (Assay ID Hs00810569_m1) and *VCAM1* (Assay ID Hs01003372_m1) was measured 48 hours after siRNA or scrambled control transfection. RNA was isolated with the miRNeasy Mini Kit (Cat. No. 217004, Qiagen, Hilden, Germany) and converted to cDNA with the High-Capacity cDNA Reverse Transcription Kit (Cat. No. 4368814, Thermo Fisher) according to the manufacturer's protocol. Real-time PCR was performed on the ViiA 7 Real-Time PCR System (Applied Biosystems, Waltham, MA) using TaqMan Gene Expression assays for all six genes (Life Technologies, Foster City, CA) with *GAPDH* as a control (Cat. No. 4448489, Assay ID Hs02786624_g1).

The ΔΔCT technique was used to calculate the relative gene expression in samples with the fold difference between the siRNA knockdown and scrambled control calculated as: fold difference = 2^(ΔΔCT). Gene expression after siRNA transfection is expressed as a percentage relative to the scrambled control expression for each gene. Four replicates were performed for the control and each individual siRNA knockdown. Two replicates were performed for the combined 3-gene knockdown. For each experiment, significance was determined by an *t* test.

## Statistics

For basic statistical analyses and when not otherwise specified, a *t* test was used to compare two continuous variables, and a Fisher's exact test was used for categorical comparisons.

## Reporting summary

Further information on research design is available in the Nature Portfolio Reporting Summary linked to this article.

# Data availability

BigWig tracks are available for all datasets in the Kidney chromatin landscape browser at https://doi.org/10.48698/HHE6-YV15. Due to privacy considerations and the nature of DNA information, raw FASTQ and BAM files are available upon request at www.kpmp.org. These files

are available without timeframe limitations. Downloadable BigWig files for whole genome bisulfite sequencing and CUT&RUN are available at https://doi.org/10.48698/HHE6-YV15. The expression matrices for the multiome are available at https://doi.org/10.48698/HHE6-YV15. The Seurat object is available in Zenodo at https://doi.org/10.5281/zenodo.8029990. All differentially expressed genes and differentially accessible peaks for all analyses are included in the supplementary data. Source data files are provided with underlying data for statistics included in figure panels. Source data are provided with this paper.

## Code availability

Code is available at https://github.com/GischD/gisch-et-al-2023.git.

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

## Acknowledgements

D.L.G. was supported by the Indiana University Grand Challenge Precision Health Initiative. M.T.E. was supported by R01AT011463-01A1, U54 DK134301-01, and U01DK114923. We would like to acknowledge the NIH Common Fund Human Biomolecular Atlas Project (HuBMAP) and the NIDDK's Kidney Precision Medicine Project. This research was supported in part by the Indiana University Pervasive Technology Institute and the Washington University Kidney Translational Research Center. MR was supported by 5U2CDK11488605 (KPMP Opportunity Pool), the Washington University Chromalloy Renal Diseases Endowment, 5R01DK12987902, and 1R21DK12879501-A1. S.J. was supported by the U54DK134301, U01DK114933, UH3DK114933, and P50DK133943. The KPMP is funded by the following grants from the NIDDK: U01DK133081, U01DK133091, U01DK133092, U01DK133093, U01DK133095, U01DK133097, U01DK114866, U01DK114908, U01DK133090, U01DK133113, U01DK133766, U01DK133768, U01DK114907, U01DK114920, U01DK114923, U01DK114933, U24DK114886.

## Author contributions

M.T.E., M.R., S.J.: conceptualization; S.J., K.J.K., A.T., J.B.H., K.K.: specimen acquisition; J.B., M.B., B.L., D.L.G., Y.H.C., B.Z., L.S., C.L.P.: data generation; Y.H.C., L.R., S.W., T.A.S., K.J.K., A.K., D.D., K.Y.C., K.S.C., A.R.S., M.J.F., D.B., M.A., S.E.R., J.P.G., S.V.P., B.H.R., I.H.D.B.: data curation; D.L.G., M.B., S.M., R.G., C.L., R.M.F., C.L.P., T.H.: formal analysis; M.T.E., M.R., S.J., P.C.D., T.M.E., J.H., M.K., K.K., J.B.H., K.Z.: funding acquisition; N.J.: patient representative; M.S.K., N.G., F.A.: creation of kidney chromatin landscape browser; all authors: investigation, methodology, and writing—review & editing; D.L.G.: project administration; D.L.G., M.B., M.T.E., M.R., S.J., S.A., J.B., S.M., R.G.: writing—original draft. K.P.M.P.: provision of infrastructure.

## Competing interests

The authors declare no competing interests.

## Additional information

Debora L. Gisch [1,53], Michelle Brennan [2,53], Blue B. Lake [3,52,53], Jeannine Basta[4,53], Mark S. Keller [5], Ricardo Melo Ferreira [1], Shreeram Akilesh[6], Reetika Ghag[4], Charles Lu[4], Ying-Hua Cheng[1], Kimberly S. Collins [1], Samir V. Parikh[7], Brad H. Rovin[7], Lynn Robbins[8], Lisa Stout[4], Kimberly Y. Conklin[3], Dinh Diep[3], Bo Zhang [4], Amanda Knoten[4], Daria Barwinska [1], Mahla Asghari[1], Angela R. Sabo [1], Michael J. Ferkowicz[1], Timothy A. Sutton [1], Katherine J. Kelly[1], Ian H. De Boer[6], Sylvia E. Rosas[9], Krzysztof Kiryluk [10], Jeffrey B. Hodgin [11], Fadhl Alakwaa[11], Seth Winfree [12], Nichole Jefferson[13], Aydın Türkmen[14], Joseph P. Gaut[3], Nils Gehlenborg[5], Carrie L. Phillips[1], Tarek M. El-Achkar[1], Pierre C. Dagher [1], Takashi Hato [1], Kun Zhang [3], Jonathan Himmelfarb[6], Matthias Kretzler[11], Shamim Mollah [4], the Kidney Precision Medicine Project (KPMP)*, Sanjay Jain [4,54] ✉, Michael Rauchman [4,54] ✉ & Michael T. Eadon [1,54] ✉

[1]Indiana University School of Medicine, Indianapolis, IN 46202, USA. [2]Saint Louis University, St. Louis, MO 63103, USA. [3]Department of Bioengineering, University of California, San Diego, La Jolla, CA, USA. [4]Washington University in Saint Louis, St. Louis, MO 63103, USA. [5]Harvard Medical School, Boston, MA 02142, USA. [6]University of Washington –, Seattle, WA 98195, USA. [7]Ohio State University Wexner Medical Center, Columbus, OH 43210, USA. [8]St. Louis Veteran Affairs Medical Center, St. Louis, MO 63106, USA. [9]Joslin Diabetes Center, Harvard Medical School, Boston, MA 02215, USA. [10]Columbia University, New York, NY 10032, USA. [11]University of Michigan, Ann Arbor, MI 48109, USA. [12]University of Nebraska Medical Center, Omaha, NE 68198, USA. [13]Kidney Precision Medicine Project Community Engagement Committee, Dallas, TX, USA. [14]Istanbul School of Medicine, Division of Nephrology, Istanbul, Turkey. [52]Present address: San Diego Institute of Science, Altos Labs, San Diego, CA, USA. [53]These authors contributed equally: Debora L. Gisch, Michelle Brennan, Blue B. Lake, Jeannine Basta. [54]These authors jointly supervised this work: Sanjay Jain, Michael Rauchman, Michael T. Eadon. *A list of authors and their affiliations appears at the end of the paper. ✉e-mail: sanjayjain@wustl.edu; mrauchma@wustl.edu; meadon@iupui.edu

## the Kidney Precision Medicine Project (KPMP)

Blue Lake[15], Alexander Morales[16], Isaac Stillman[16], Stewart Lecker[16], Steve Bogen[17], Ashish Verma[18], Guanghao Yu[18], Insa Schmidt[18], Joel Henderson[18], Laurence Beck[18], Pranav Yadati[18], Sushrut Waikar[18], Afolarin A. Amodu[18], Shana Maikhor[18], Titlayo Ilori[18], Mia R. Colona[19], Astrid Weins[19], Gearoid McMahon[19], Nir Hacohen[20], Anna Greka[20], Jamie L. Marshall[20], Paul J. Hoover[20], Vidya S. Viswanathan[21], Dana Crawford[21], Mark Aulisio[21], William Bush[21], Yijiang Chen[21], Anant Madabhushi[21], Charles O'Malley[22], Crystal Gadegbeku[22], Dianna Sendrey[22], Emilio Poggio[22], John O'Toole[22], John Sedor[22], Jonathan Taliercio[22], Lakeshia Bush[22], Leal Herlitz[22], Ellen Palmer[22], Jane Nguyen[22], Kassandra Spates-Harden[22], Leslie Cooperman[22], Stacey Jolly[22], Carissa Vinovskis[23], Andrew Bomback[24], Jonathan Barasch[24], Krzysztof Kiryluk[24], Paul Appelbaum[24], Vivette D'Agati[24], Cecilia Berrouet[24], Karla Mehl[24], Maya Sabatello[24], Ning Shang[24], Olivia Balderes[24], Pietro A. Canetta[24], Satoru Kudose[24], Joana de Pinho Gonçalves[25], Lukasz Migas[25], Raf Van de Plas[25], Roy Lardenoije[25], Laura Barisoni[26], Helmut Rennke[5], Abraham Verdoes[1], Angela Sabo[1], Daria Barwinska [1], Debora Lidia Gisch[1], James Williams[1], Katherine Kelly[1], Kenneth Dunn[1], Mahla Asghari[1], Michael Eadon[1], Michael Ferkowicz[1], Pierre Dagher[1], Ricardo Melo Ferreira [1], Seth Winfree[1], Sharon Bledsoe[1], Stephanie Wofford[1], Tarek M. El-Achkar[1], Timothy Sutton[1], William Bowen[1], Ying-Hua Cheng[1], Austen Slade[1], Elizabeth Record[1], Yinghua Cheng[1], Yashvardhan Jain[1], Bruce Herr[27], Ellen Quardokus[27], Ashley Wang[28], Celia Pamela Corona Villalobos[28], Chirag Parikh[28], Mohamed Atta[28], Steven Menez[28], Yumeng Wen[28], Alan Xu[28], Lauren Bernard[28], Camille Johansen[9], Sarah Chen[9], Sylvia Rosas[9], Isabel Donohoe[9], Jennifer Sun[9], Richard Knight[29], Anna Shpigel[29], Jack Bebiak[29], John Saul[29], Joseph Ardayfio[29], Robert Koewler[29], Roy Pinkeney[29], Taneisha Campbell[29], Evren Azeloglu[30], Girish Nadkarni[30], John He[30], Joji Tokita[30], Kirk Campbell[30], Marissa Patel[30], Sean Lefferts[30], Srinivas Ravi Iyengar[30], Stephen Ward[30],

Steven Coca[30], Cijang He[30], Yuguang Xiong[30], Pottumarthi Prasad[31], Brad Rovin[7], John P. Shapiro[7], Samir Parikh[7], Sethu M. Madhavan[7], Jessica Lukowski[32], Dusan Velickovic[32], Ljiljana Pasa-Tolic[32], George (Holt) Oliver[33], Olga Troyanskaya[34], Rachel Sealfon[34], Weiguang Mao[34], Aaron Wong[34], Ari Pollack[35], Yury Goltsev[36], Brandon Ginley[37], Brendon Lutnick[37], Kun Zhang[3], Garry Nolan[38], Kavya Anjani[38], Tariq Mukatash[38], Zoltan G. Laszik[38], Baltazar Campos[39], Bijin Thajudeen[39], David Beyda[39], Erika Bracamonte[39], Frank Brosius[39], Gregory Woodhead[39], Katherine Mendoza[39], Nicole Marquez[39], Raymond Scott[39], Rebecca Tsosie[39], Milda Saunders[40], Adele Rike[41], E. Steve Woodle[41], Paul J. Lee[41], Rita R. Alloway[41], Tiffany Shi[41], Elena Hsieh[23], Jessica Kendrick[23], Joshua Thurman[23], Julia Wrobel[23], Laura Pyle[23], Petter Bjornstad[23], Nicholas Lucarelli[42], Pinaki Sarder[42], Amada Renteria[43], Ana Ricardo[43], Anand Srivastava[43], Devona Redmond[43], Eunice Carmona-Powell[43], James Bui[43], James Lash[43], Monica Fox[43], Natalie Meza[43], Ron Gaba[43], Suman Setty[43], Tanika Kelly[43], Chrysta Lienczewski[11], Dawit Demeke[11], Edgar Otto[11], Fadhl Alakwaa[11], Heather Ascani[11], Jeff Hodgin[11], Jennifer Schaub[11], John Hartman[11], Laura Mariani[11], Markus Bitzer[11], Matthias Kretzler[11], Michael Rose[11], Nikki Bonevich[11], Ninive Conser[11], Phillip McCown[11], Rachel Dull[11], Rajasree Menon[11], Rebecca Reamy[11], Sean Eddy[11], Ul Balis[11], Victoria Blanc[11], Viji Nair[11], Yongqun Oliver He[11], Zachary Wright[11], Becky Steck[11], Jinghui Luo[11], Renee Frey[11], Alyson Coleman[44], Dorisann Henderson-Brown[44], Jerica Berge[44], Maria Luiza Caramori[44], Oyedele Adeyi[44], Patrick Nachman[44], Sami Safadi[44], Siobhan Flanagan[44], Sisi Ma[44], Susan Klett[44], Susan Wolf[44], Tasma Harindhanavudhi[44], Via Rao[44], Amy Mottl[45], Anne Froment[45], Evan Zeitler[45], Peter Bream[45], Sara Kelley[45], Matthew Rosengart[46], Michele Elder[46], Paul Palevsky[46], Raghavan Murugan[46], Daniel E. Hall[46], Filitsa Bender[46], James Winters[46], John A. Kellum[46], Matthew Gilliam[46], Mitchell Tublin[46], Roderick Tan[46], Guanshi Zhang[47], Kumar Sharma[47], Manjeri Venkatachalam[47], Allen Hendricks[48], Asra Kermani[48], Jose Torrealba[48], Miguel Vazquez[48], Nancy Wang[48], Qi Cai[48], Richard Tyler Miller[48], Shihong Ma[48], Susan Hedayati[48], Andrew Hoofnagle[6], Artit Wangperawong[6], Ashley Berglund[6], Ashveena L. Dighe[6], Bessie Young[6], Brandon Larson[6], Brooke Berry[6], Charles Alpers[6], Christine Limonte[6], Christy Stutzke[6], Glenda Roberts[6], Ian de Boer[6], Jaime Snyder[6], Jimmy Phuong[6], Jonas Carson[6], Jonathan Himmelfarb[6], Kasra Rezaei[6], Katherine Tuttle[6], Keith Brown[6], Kristina Blank[6], Natalya Sarkisova[6], Nichole Jefferson[6], Robyn McClelland[6], Sean Mooney[6], Yunbi Nam[6], Adam Wilcox[6], Christopher Park[6], Frederick Dowd[6], Kayleen Williams[6], Stephanie M. Grewenow[6], Stephen Daniel[6], Stuart Shankland[6], Annapurna Pamreddy[49], Hongping Ye[49], Richard Montellano[49], Shweta Bansal[49], Anil Pillai[48], Dianbo Zhang[48], Harold Park[48], Jiten Patel[48], Kamalanathan Sambandam[48], Mujeeb Basit[48], Natasha Wen[48], Orson W. Moe[48], Robert D. Toto[48], Simon C. Lee[48], Kavya Sharman[50], Richard M. Caprioli[50], Agnes Fogo[50], Jamie Allen[50], Jeffrey Spraggins[50], Katerina Djambazova[50], Mark de Caestecker[50], Martin Dufresne[50], Melissa Farrow[50], Amanda Knoten[4], Anitha Vijayan[4], Bo Zhang[4], Brittany Minor[4], Gerald Nwanne[4], Jeannine Basta[4,53], Joseph Gaut[4], Kristine Conlon[4], Madhurima Kaushal[4], Sanjay Jain[4,54] ✉, Sabine M. Diettman[4], Angela M. Victoria Castro[51], Dennis Moledina[51], Francis P. Wilson[51], Gilbert Moeckel[51], Lloyd Cantley[51], Melissa Shaw[51], Vijayakumar Kakade[51] & Tanima Arora[51]

[15]Altos Labs, San Diego, CA, USA. [16]Beth Israel Deaconess Medical Center, Boston, USA. [17]Boston Cell Standards, Boston, MA, USA. [18]Boston Medical Center, Boston, MA, USA. [19]Brigham & Women's Hospital, Boston, USA. [20]Broad Institute, Cambridge, MA, USA. [21]Case Western Reserve, Cleveland, OH, USA. [22]Cleveland Clinic, Cleveland, OH, USA. [23]University of Colorado, Denver, CO, USA. [24]Columbia University, New York, NY, USA. [25]Delft University of Technology, Delft, Netherlands. [26]Duke University, Durham, NC, USA. [27]Indiana University Bloomington, Bloomington, IN, USA. [28]Johns Hopkins University, Baltimore, MD, USA. [29]KPMP Patient Partner, New York, NY, USA. [30]Mount Sinai, New York, NY, USA. [31]Northwestern University, Chicago, IL, USA. [32]Pacific Northwest National Laboratories, Richland, WA, USA. [33]PCCI, Southlake, TX, USA. [34]Princeton University, Princeton, NJ, USA. [35]Seattle Children's Hospital, Seattle, WA, USA. [36]Stanford University, Stanford, CA, USA. [37]SUNY Buffalo, Buffalo, NY, USA. [38]UC San Francisco, Seattle, WA, USA. [39]University of Arizona, Tucson, AZ, USA. [40]University of Chicago, Chicago, IL, USA. [41]University of Cincinnati, Cincinnati, OH, USA. [42]University of Florida, Gainsville, USA. [43]University of Illinois, Chicago, IL, USA. [44]University of Minnesota, Minneapolis, MN, USA. [45]University of North Carolina, Chapel Hill, NC, USA. [46]University of Pittsburgh, Pittsburgh, PA, USA. [47]University of Texas Health Science Center at San Antonio, San Antonio, TX, USA. [48]University of Texas Southwestern, Dallas, TX, USA. [49]UT Health San Antonio, San Antonio, TX, USA. [50]Vanderbilt University, Nashville, TN, US. [51]Yale University, New Haven, CT, USA.

