## [Peer Review File · Nature Communications]

The chromatin landscape of healthy and injured cell types in the human kidneyREVIEWER COMMENTS

Reviewer #1 (Remarks to the Author):

In the current manuscript, Gish et al. evaluated a set of human kidney tissue samples with multiple approaches to establish an epigenome atlas. Based on the postulate that there is a need to define regions of gene activation or repression that control human kidney cells in states of health, injury, and repair, the authors propose the resulting epigenome kidney atlas as the foundation to enable targeted cell-specific therapeutic interventions by reprogramming gene regulatory networks.

Adaptive states with successful or maladaptive characteristics were recently described in both the proximal tubule (PT) and the thick ascending limb (TAL) epithelial cells of the loop of Henle (preprint available data, <https://www.biorxiv.org/content/10.1101/2021.07.28.454201v1>) that is applied for integration and elaboration of results in the present manuscript.

The main results and findings from the current manuscript include:

- 1) The authors provide extensive information supporting the quality of the data and show scientific rigor in experimental evaluations.
- 2) Using an integrative analysis combining bulk tissue ATAC and cortex CUT&RUN histone data with region-specific DNA methylation, the results support the capability to use bulk level data to augment region-specific data and gain additional insights into region-specific epigenetic landscape information.
- 3) Cell type-specific open chromatin signature across the kidney's varied range of cell types was done using multiome (snRNAseq & snATACseq) in samples from 8 kidneys (7 nephrectomies and 1 biopsy) for a total of 12 samples (47,217 nuclei). 4 out of 8 kidneys with multiome data overlapped WGBS DNAm and one out of 8 also had CUT & RUN data (from Supplemental Table 1). The multiome data yielded 72 distinct cell types and the authors reported agreement among technologies as well as with previous reports (doi:10.1038/s41467-021-22368-w). Also, the data supported that the multiome data set provides the resolution to identify PT subtypes.
- 4) The authors demonstrated genome-wide orthogonal validation between complementary epigenomic features.
- 5) The authors reported a proximal tubule cell transcription factor network of ELF3, KLF6,

and KLF10 regulated the transition between health and injury. In addition, combined perturbation of ELF3, KLF6, and KLF10 distinguished two adaptive proximal tubular cell subtypes, one of which manifested a repair trajectory after knockout (using CellOracle in silico knockout).

6) The study includes spatial localization using spatial transcriptomics.

Some concerns were identified including:

1) Patients and samples. Supplemental Table 1 shows the characteristics of patients, samples, and approaches used for individual samples. Overall, some of the information provided as part of this Table is not clear, and the questions/commentaries below deserve further attention:

a) The authors state that to understand the contributions of epigenomic features to transcript expression regulation, they generated, integrated, and aligned data from multiple orthogonal technologies performed on a partially overlapping set of 23 samples (Page #5, under: "Samples and Quality Control"). However, it looks like 23 kidneys were evaluated using multiple samples from some of them. This needs to be clarified.

b) Instead of 23 kidneys, a total of 25 are listed, with one of them (clinical diagnosis of AKI) only tested using spatial transcriptomics and regional mRNA and a second only using spatial transcriptomics (clinical diagnosis DKD). It seems that the 23 kidneys are the ones evaluated using epigenetic evaluations. However, these discrepancies need to be further explained to improve clarity.

c) The authors assert "Although this epigenomic atlas does not include diseased kidney biopsy samples, both the reference and adaptive cell states are found ubiquitously in nephrectomy samples, which have a modicum of chronic injury" (Page#16). However, sample 32-10074 is a biopsy with AKI and sample s1815183 is a biopsy with a clinical diagnosis of FSGS (diseased kidneys) and are listed as evaluated with WGBS DNAm. Were these samples included in the analyses? Overall, this statement is confusing based on current Table 1 and needs further clarification.

d) It seems like the patient's Race is not included as part of the Table. This information is relevant and needs to be included. This is important not only to balance clinical variable but also because some races can relate to specific alternations in genes (e.g., APOL1 in AA) that associate with higher risk of disease.

e) DNAm relates to aging, however, there is no mention of a possible role of age in findings.

The age range for patients whose kidneys were evaluated using WGBS is 20-79 yo. This point needs further evaluation.

f) The first 3 samples listed in the Table and used for multiome do not have a reported clinical diagnosis and the histological reports look considerably different. Overall, the sample set is considerably histologically heterogeneous, and there is no discussion about the influence of the etiology of renal disease (e.g., FSGS, AKI, IFTA) and other important variables (e.g., ischemia time for donor nephrectomy kidneys reported as influencing DNAm patterns doi: 10.1681/ASN.2017091027) on epigenetic patterns.

2) Multiome: Although the authors report the identification of 72 cell clusters, information about the distribution of these clusters among samples is not provided. Did the authors identify cell clusters unique for some samples? If so, how do the clusters relate to histological evidence of chronic injury or other histological features?

3) Supporting validations for some of the key computational findings in the study would have significantly increased the impact of the study. As an example, this is directly related to the validation of the results from CellOracle in silico knockout approaches.

4) Discussion statements: The Discussion mentions “Because our work builds upon a transcriptomic atlas of healthy and diseased tissue, we were able to explore the regulatory network of the adaptive cell state that associated with progression of CKD and AKI to CKD transition. Although this epigenomic atlas does not include diseased kidney biopsy samples, both the reference and adaptive cell states are found ubiquitously in nephrectomy samples, which have a modicum of chronic injury.” This statement confusing as based on this conclusion, samples like 17-1606 (Focal global glomerulosclerosis 30%. 40% IFTA. Mild arteriosclerosis. Mild chronic interstitial inflammation) are considered as “non-diseased kidney”. The concept of normal or healthy for sample classification is complicated, as healthy tissue is quite heterogenous and it has been reported that “healthy reference” can contain abnormal pathology, cells, and/or molecular signatures. However, some discussion about this point and the criteria for selection of the samples used for the studies would be extremely useful for the readers (especially when the authors emphasized the not use “of diseased kidneys” in epigenetic evaluations in the present study and encouraged the use of diseased kidneys in future studies).

Minor:

- Track sizes of snATAC-seq are small and difficult to interpret.

- Can the authors please confirm that the deceased donor kidney (sample ID 3409) with a histological report of 46% focal global glomerulosclerosis, mild IFTA, and mild chronic interstitial inflammation and arteriosclerosis has a clinical diagnosis of “tumor nephrectomy”.
- Overall, the Figures have too many panels, with many of them being small and not easy to read.
- Supplemental information will benefit for additional clarification. As examples, Table 1 does not include meaning of abbreviations, and Table 2 does not clarify if the Table include CpGs or DMRs.

Reviewer #2 (Remarks to the Author):

Thank you to the team for this very large body of Omics and bioinformatics research. This is a significant study, with some new and interesting information. Specifically, the authors demonstrate a transcription factor network that that is key for regulating novel chromatin peaks in adaptive proximal tubules. They provide an in silico analysis how deletion of these transcription factors would change the fate of an adapting proximal tubule (they also go and do the same type of analysis for adapting thick ascending limbs).

I have a few minor suggestions.

1) Although I appreciate why you focused on PT and TAL, is it possible to provide any information about other cell types in your samples? Can you at least provide a conical marker for each other cell type in the supplement, similar to figure 2A? Aqp2, Slc12a3, Pecam1, etc? I realize you have 72 cell types, but maybe you can at least report for the epithelial cells and endothelial ones. If that really isn't feasible, will your online resource have that capability, so a reader could look at the principal cells for example?

2) Figure 2A in the results was a bit hard to follow when you mention the other peaks in introns and the final exon. Can you highlight those peaks in grey (or another color) for clarity because I didn't see the DNA methylation "dips" for these other sites like mentioned.

3) will you code used for analysis and figure generation be deposited anywhere?

Reviewer #1 (Remarks to the Author):

In the current manuscript, Gish et al. evaluated a set of human kidney tissue samples with multiple approaches to establish an epigenome atlas. Based on the postulate that there is a need to define regions of gene activation or repression that control human kidney cells in states of health, injury, and repair, the authors propose the resulting epigenome kidney atlas as the foundation to enable targeted cell-specific therapeutic interventions by reprogramming gene regulatory networks.

Adaptive states with successful or maladaptive characteristics were recently described in both the proximal tubule (PT) and the thick ascending limb (TAL) epithelial cells of the loop of Henle (preprint available data, <https://www.biorxiv.org/content/10.1101/2021.07.28.454201v1>) that is applied for integration and elaboration of results in the present manuscript.

The main results and findings from the current manuscript include:

- 1) The authors provide extensive information supporting the quality of the data and show scientific rigor in experimental evaluations.
- 2) Using an integrative analysis combining bulk tissue ATAC and cortex CUT&RUN histone data with region-specific DNA methylation, the results support the capability to use bulk level data to augment region-specific data and gain additional insights into region-specific epigenetic landscape information.
- 3) Cell type-specific open chromatin signature across the kidney's varied range of cell types was done using multiome (snRNAseq & snATACseq) in samples from 8 kidneys (7 nephrectomies and 1 biopsy) for a total of 12 samples (47,217 nuclei). 4 out of 8 kidneys with multiome data overlapped WGBS DNAm and one out of 8 also had CUT & RUN data (from Supplemental Table 1). The multiome data yielded 72 distinct cell types and the authors reported agreement among technologies as well as with previous reports (doi:10.1038/s41467-021-22368-w). Also, the data supported that the multiome data set provides the resolution to identify PT subtypes.
- 4) The authors demonstrated genome-wide orthogonal validation between complementary epigenomic features.
- 5) The authors reported a proximal tubule cell transcription factor network of ELF3, KLF6, and KLF10 regulated the transition between health and injury. In addition, combined perturbation of ELF3, KLF6, and KLF10 distinguished two adaptive proximal tubular cell subtypes, one of which manifested a repair trajectory after knockout (using CellOracle in silico knockout).
- 6) The study includes spatial localization using spatial transcriptomics.

Thank you for your thorough read of the manuscript and constructive comments.

Some concerns were identified including:

- 1) Patients and samples. Supplemental Table 1 shows the characteristics of patients, samples, and approaches used for individual samples. Overall, some of the information provided as part of this Table is not clear, and the questions/commentaries below deserve further attention:
 - a) The authors state that to understand the contributions of epigenomic features to transcript expression regulation, they generated, integrated, and aligned data from multiple orthogonal technologies performed on a partially overlapping set of 23 samples (Page #5, under: "Samples and Quality Control"). However, it looks like 23 kidneys were evaluated using multiple samples from some of them. This needs to be clarified.

Thank you for this comment. We have updated the total number of specimens to be 25, with 23 unique epigenetic specimens. Multiple replicates and multiple technologies were run for some samples resulting in a total of 84 datasets.

- b) Instead of 23 kidneys, a total of 25 are listed, with one of them (clinical diagnosis of AKI) only tested using spatial transcriptomics and regional mRNA and a second only using spatial transcriptomics (clinical

diagnosis DKD). It seems that the 23 kidneys are the ones evaluated using epigenetic evaluations. However, these discrepancies need to be further explained to improve clarity.

Thank you. We have clarified this as indicated above.

c) The authors assert “Although this epigenomic atlas does not include diseased kidney biopsy samples, both the reference and adaptive cell states are found ubiquitously in nephrectomy samples, which have a modicum of chronic injury” (Page#16). However, sample 32-10074 is a biopsy with AKI and sample s1815183 is a biopsy with a clinical diagnosis of FSGS (diseased kidneys) and are listed as evaluated with WGBS DNAm. Were these samples included in the analyses? Overall, this statement is confusing based on current Table 1 and needs further clarification.

This is a good point. These samples were included in the analysis. As you suggested, we have edited this statement to enhance clarity:

Although this epigenomic atlas includes a limited number of diseased kidney biopsy samples, both the reference and adaptive cell states are found in nephrectomy and deceased donor samples, which have varying degrees of injury. Healthy tissue is heterogenous and often contains abnormal pathology and altered cells with injury signatures, which could be attributed to factors including physiological aging, comorbidities, or procedure-related changes. The overlap between cells with healthy and injury cell states may serve as a bridge between reference and disease samples and allowed us to infer regulatory factors in these states that are likely to be applicable to disease samples. These regulatory relationships will be further explored as larger cohorts are analyzed by these technologies in the Kidney Precision Medicine Project and HuBMAP.

d) It seems like the patient’s Race is not included as part of the Table. This information is relevant and needs to be included. This is important not only to balance clinical variable but also because some races can relate to specific alternations in genes (e.g., APOL1 in AA) that associate with higher risk of disease.

We agree that race is an important variable. We have included race in Supplemental Table 1 when that information was available.

e) DNAm relates to aging, however, there is no mention of a possible role of age in findings. The age range for patients whose kidneys were evaluated using WGBS is 20-79 yo. This point needs further evaluation.

This is a good point. We have conducted an analysis to examine the relationship between age and methylation. We examined all genes in the glomerulus and tubulointerstitium. We also examined the subset of markers genes for the adaptive proximal tubule (aPT). No association was found. Overall, we are underpowered at our sample size. There are only 2 individuals under the age of 40. Your suggestion would make an important follow-up analysis when more sample data is available. We have included these data here for the reviewers but did not add it to the manuscript given the underpowered nature of the analysis at this time.

Here is the correlation of age with whole genome methylation:

- Methylation levels GLOM by age

- Methylation levels TI by age

- Subset of genes upregulated in aPT cells.
 - Methylation levels in TI by sample, ordered by age

f) The first 3 samples listed in the Table and used for multiome do not have a reported clinical diagnosis and the histological reports look considerably different. Overall, the sample set is considerably histologically heterogeneous, and there is no discussion about the influence of the etiology of renal disease (e.g., FSGS, AKI, IFTA) and other important variables (e.g., ischemia time for donor nephrectomy kidneys reported as influencing DNAm patterns doi: 10.1681/ASN.2017091027) on epigenetic patterns.

Thank you for pointing this out. We have added a clinical diagnosis to the 3 samples in Supplemental Table 1. We included your suggested reference in the discussion and a brief discussion of the effects of renal disease on epigenetics.

2) Multiome: Although the authors report the identification of 72 cell clusters, information about the distribution of these clusters among samples is not provided. Did the authors identify cell clusters unique for some samples? If so, how do the clusters relate to histological evidence of chronic injury or other histological features?

Thank you. We ran an additional analysis for the cell type distribution across specimens (Supplemental Figure 4i). We found some variation across samples. For example, the sample 21-020 had a greater proportion of medullary cells and was 50% medulla on histology. Sample 3691 had greater B cell representation and had interstitial infiltrate on histopathologic exam. The main cell types of interest (PT-S1, PT-S2, and aPT) for this study were well represented across the samples. The sample size precluded statistical comparisons across histological features, but this is a major focus of the KPMP over the next several years. Thus, we focused our study on aligning diverse datasets and looking for commonalities across the genome.

3) Supporting validations for some of the key computational findings in the study would have significantly increased the impact of the study. As an example, this is directly related to the validation of the results from CellOracle in silico knockout approaches.

Thank you for this critique. To provide wet lab validation, we performed several siRNA knockdown experiments in NHPTK cells (normal human proximal tubular kidney cells). These cells were immortalized by hTERT elongation and at passage 7. We conducted knockdown of *ELF3*, *KLF10*, *KLF6*, and a combination of the three and evaluated target gene expression of *TPM1*, *ITGB8*, and *VCAM1*. Knockdown of *ELF3* and *KLF10* led to reduced expression of *TPM1*, *ITGB8*, and *VCAM1*. These data support the in silico knockdown results and were added to Supplemental Figure 11.

4) Discussion statements: The Discussion mentions “Because our work builds upon a transcriptomic atlas of healthy and diseased tissue, we were able to explore the regulatory network of the adaptive cell state that associated with progression of CKD and AKI to CKD transition. Although this epigenomic atlas does not include diseased kidney biopsy samples, both the reference and adaptive cell states are found ubiquitously in nephrectomy samples, which have a modicum of chronic injury.” This statement confusing as based on this conclusion, samples like 17-1606 (Focal global glomerulosclerosis 30%. 40% IFTA. Mild arteriosclerosis. Mild chronic interstitial inflammation) are considered as “non-diseased kidney”. The concept of normal or healthy for sample classification is complicated, as healthy tissue is quite heterogenous and it has been reported that “healthy reference” can contain abnormal pathology, cells, and/or molecular signatures. However, some discussion about this point and the criteria for selection of the samples used for the studies would be extremely useful for the readers (especially when the authors emphasized the not use “of diseased kidneys” in epigenetic evaluations in the present study and encouraged the use of diseased kidneys in future studies).

Thank you for this point. We have updated this statement as discussed above in item 1c.

Minor:

- Track sizes of snATAC-seq are small and difficult to interpret.

Thank you for this comment. We recognize there is a large amount of data which is displayed across the whole genome or a whole gene. We will work with the Nature editorial staff to further optimize the display as required. Specifically, we made the following changes:

1. Figure 4 was split in half to make Fig 4 and 5.
2. Figure 5 (now Fig 6)– The ITGB3 tracks and panel F feature plots were moved to supplemental figure 10. The remaining tracks for TPM1 and KLF6 were enlarged.
3. Figure 7 (now Fig 8)– The gene tracks for TM4SF1 were moved to Supplemental Figure 11. The remaining panels were enlarged.

- Can the authors please confirm that the deceased donor kidney (sample ID 3409) with a histological report of 46% focal global glomerulosclerosis, mild IFTA, and mild chronic interstitial inflammation and arteriosclerosis has a clinical diagnosis of “tumor nephrectomy”.

Good point. We recategorized this sample as tumor nephrectomy with CKD. All samples with moderate (26%) or greater glomerular obsolescence or IFTA were recategorized as a nephrectomy with CKD.

- Overall, the Figures have too many panels, with many of them being small and not easy to read.

Thanks. We have made the edits as above and split selected figures to enhance readability.

- Supplemental information will benefit for additional clarification. As examples, Table 1 does not include meaning of abbreviations, and Table 2 does not clarify if the Table include CpGs or DMRs.

Thank you. We have added the abbreviations to Supplemental Table 1. For Supplemental Table 2, we clarified that it contains summative methylation across gene regions.

Reviewer #2 (Remarks to the Author):

Thank you to the team for this very large body of Omics and bioinformatics research. This is a significant study, with some new and interesting information. Specifically, the authors demonstrate a transcription factor network that is key for regulating novel chromatin peaks in adaptive proximal tubules. They provide an in silico analysis how deletion of these transcription factors would change the fate of an adapting proximal tubule (they also go and do the same type of analysis for adapting thick ascending limbs).

Thank you for these comments and assessment of the manuscript.

I have a few minor suggestions.

1) Although I appreciate why you focused on PT and TAL, is it possible to provide any information about other cell types in your samples? Can you at least provide a conical marker for each other cell type in the supplement, similar to figure 2A? Aqp2, Slc12a3, Pecam1, etc? I realize you have 72 cell types, but maybe you can at least report for the epithelial cells and endothelial ones. If that really isn't feasible, will your online resource have that capability, so a reader could look at the principal cells for example?

Thank you for this suggestion. To respond, we conducted an additional analysis (now Supplemental figure 8) which includes aligned datasets for SLC12A3 (distal convoluted tubule), AQP2 (collecting duct), and PECAM1 (endothelial cell). The Multiome/ATAC-seq tracks show the expected differences in expression and open chromatin in appropriate cell types. For SLC12A3 and AQP2, the DCT and collecting duct make up a smaller proportion of the bulk sample. Thus, the DNA methylation and CUT&RUN data signals are not as pronounced due to their bulk nature.

2) Figure 2A in the results was a bit hard to follow when you mention the other peaks in introns and the final exon. Can you highlight those peaks in grey (or another color) for clarity because I didn't see the DNA methylation "dips" for these other sites like mentioned.

We added additional annotation in Fig 2A with gray stripes to denote these features.

3) will you code used for analysis and figure generation be deposited anywhere?

We have now added all our code under the associated GITHUB address:

<https://github.iu.edu/dgisch/gisch-et-al-2023>

REVIEWERS' COMMENTS

Reviewer #1 (Remarks to the Author):

The authors were very responsive to my concerns, and I do not have additional comments. The revised manuscript has been improved and present important data for the readers and moving the field forward.

Reviewer #2 (Remarks to the Author):

Thank you for addressing my previous comments. I have no further suggestions.